# Distilling LLM Prior to Flow Model for Generalizable Agent's Imagination in Object Goal Navigation

**Badi Li**[1,2]**, Ren-Jie Lu**[1]**, Yu Zhou**[1]**, Jingke Meng**[1]***, Wei-Shi Zheng**[1,3]
[1]Sun Yat-sen University [2] The University of Hong Kong
[3]Key Laboratory of Machine Intelligence and Advanced Computing, Ministry of Education
`badi.li.cs@connect.hku.hk, mengjke@gmail.com, wszheng@ieee.org`

## Abstract

The Object Goal Navigation (ObjectNav) task challenges agents to locate a specified object in an unseen environment by imagining unobserved regions of the scene. Prior approaches rely on deterministic and discriminative models to complete semantic maps, overlooking the inherent uncertainty in indoor layouts and limiting their ability to generalize to unseen environments. In this work, we propose GOAL, a generative flow-based framework that models the semantic distribution of indoor environments by bridging observed regions with LLM-enriched full-scene semantic maps. During training, spatial priors inferred from large language models (LLMs) are encoded as two-dimensional Gaussian fields and injected into target maps, distilling rich contextual knowledge into the flow model and enabling more generalizable completions. Extensive experiments demonstrate that GOAL achieves state-of-the-art performance on MP3D and Gibson, and shows strong generalization in transfer settings to HM3D. Codes and pretrained models are available at `https://github.com/Badi-Li/GOAL`.

## 1 Introduction

Embodied navigation [2, 22, 30, 43, 56, 70], which enables agents to move purposefully through complex, realistic environments, is a fundamental challenge in embodied intelligence. Within this domain, Object Goal Navigation (ObjectNav) tasks an agent with locating an instance of a user-specified object category (e.g., "find a chair") in an unseen environment, relying solely on visual observations.

To succeed at ObjectNav, the agent must not only recognize the goal object when it becomes visible but also infer its likely location before it is seen. This imagination step is particularly challenging, as it requires reasoning about contextual and co-occurrence relationships between objects (e.g., chairs often appear near tables). Recent approaches [17, 25, 75] address this by incrementally constructing top-down semantic maps and predicting full-scene semantic maps through discriminative and deterministic models. However, their deterministic nature, which directly maps inputs to fixed outputs with a strict one-to-one mapping, inherently limits generalization to unseen data.

In contrast, we argue that semantic map completion is inherently uncertain: Multiple plausible full scenes can correspond to the same partial map, and multimodal outputs could benefit generalization capabilities [10]. We therefore formulate this task as a probabilistic generation problem, leveraging recent advances in flow-based generative modeling [21, 26, 28, 53, 54] to learn the semantic distribution of indoor scenes (illustrated in Fig. 1). However, while generative models offer better generalizability, we find that applying generative models to ObjectNav poses three core challenges, which we address in this work.

---

* Corresponding author.

39th Conference on Neural Information Processing Systems (NeurIPS 2025).

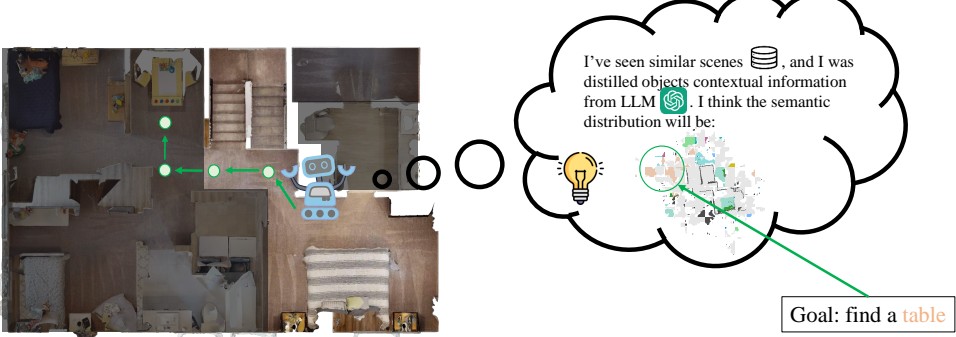

Figure 1: We incorporate a flow model to generate the semantic distribution of unobserved regions (in dark), based on dataset-internal patterns and external knowledge from language models.

First, generative models typically require large and diverse datasets to effectively learn the latent distribution. However, existing indoor scene datasets remain limited in both scale and variety. To overcome this, we incorporate external knowledge from large language models (LLMs) by *firstly modeling it as a natural Gaussian distribution in the latent space of the flow model, which enriches contextual signals during training*. In particular, we prompt LLMs to infer spatial contextual priors, including common distances between object pairs and associated confidence scores. These distance-confidence pairs are transformed into two-dimensional Gaussian priors and injected into semantic maps during training. This process distills the rich contextual information from LLM to our generative flow model, enhancing its understanding of object co-occurrence and spatial context. Crucially, this external supervision is applied only during training, avoiding inference-time costs such as API latency and memory overhead, and enabling the flow model to operate as a plug-and-play semantic reasoner.

Second, we identify the problem of inefficient conditioning. Traditional diffusion models assume the source distribution to be standard Gaussian and develop their theory based on this assumption. As a result, conditional generation in the diffusion literature typically relies on additional mechanisms, such as cross-attention, concatenation, or FiLM [38], to incorporate conditioning information. While these mechanisms are acceptable for image generation and restoration, they introduce additional computation, which may be prohibitive in visual navigation tasks that require multiple inferences per episode and real-time interaction. In contrast, the Flow Matching algorithm does not strictly assume a Gaussian source distribution, allowing us to design a more efficient conditioning mechanism: *directly modeling the dependent couplings between noise injected partial semantci maps and full LLM-enriched semantic maps*.

Third, semantic maps constructed during navigation are prone to accumulating errors from upstream segmentation models, which can degrade the performance of generative models. To mitigate this, *we aggregate past RGB-D observations into unified point clouds representations and perform joint segmentation using 3D perception models*, inspired by how humans implicitly integrate multi-frame observations. This method captures both spatial geometry and temporal consistency more effectively than traditional 2D semantic segmentation methods used in ObjectNav [20, 23]. As a result, we achieve more accurate and consistent scene understanding.

In conclusion, we propose GOAL (**G**uiding Agent's imaginati**O**n with gener**A**ive f**L**ow), a generative framework that incorporates external knowledge as training supervision, models direct couplings to better leverage semantic map priors, and integrates multi-view observations for enhanced scene-level understanding. These components together lead to strong and generalizable performance on the ObjectNav task.

Evaluations on large-scale datasets Gibson[63] and MP3D [8] show that our approach significantly outperforms baselines. Additionally, transfer experiments, training on MP3D and testing on HM3D [41], demonstrate the strong generalization capabilities of our approach to unseen environments.

## 2 Related Work

**ObjectGoal navigation.** ObjectNav approaches fall into two main categories: end-to-end and modular. End-to-end methods map visual inputs to actions using reinforcement or imitation learning,

focusing on improving visual representations [7, 64, 65, 66] or tackling policy learning challenges like sparse rewards and overfitting [42, 50, 59, 67, 68]. Modular methods decompose the task into components such as mapping, planning, and policy learning. Given a semantic map, these methods explore waypoint or frontier selection [9, 32, 40], distance estimation [79], target probability prediction [73], and semantic map completion [17, 25, 75]. T-Diff [72] firstly introduced generative modeling to ObjectNav via a DDPM conditioned on semantic maps for trajectory generation. Alternatively , we propose a generative flow-based model that imagines full-scene semantics, using LLM-derived priors to improve generalization to unseen environments.

**Diffusion and flow-based generative models.** Generative models such as diffusion [21], and flow-based methods [26, 28] generate data via iterative denoising or learned velocity fields. They have shown strong performance in image generation and restoration tasks [31, 45, 47, 48, 78]. We draw an analogy between agent's imagination via completing a partial semantic map and image restoration, where the goal is to reconstruct missing content from degraded input. Most approaches begin from Gaussian noise and condition on the input via concatenation, FiLM [38], or cross-attention. Recent alternatives have explored directly bridging degraded images and targets via Schrödinger Bridges [27] and stochastic interpolants [1] , but often trade off quality for interpretability. In contrast, we show that for sparse semantic maps, direct coupling via flow matching can be more efficient without sacrificing the performance. We adopt flow matching framework for its faster sampling and mathematically simple yet expressive formulation.

## 3 Flow Matching Preliminaries

The generative task is typically defined as a mapping $\psi_t$, $t \in [0, 1]$, which transports samples $X_0$ from a source data distribution $p$ to samples $X_1$ in a target distribution $q$. In general, the source and target samples may come from a joint distribution $(X_0, X_1) \sim \pi_{0,1}(X_0, X_1)$.

To address this transformation from $X_0$ to $X_1$, Flow Matching [26] interpolates a probability path $p_t$ between the source and target samples:

$$X_t = \alpha_t X_0 + \beta_t X_1 \sim p_t, \tag{1}$$

with boundary conditions $\alpha_0 = \beta_1 = 1$ and $\alpha_1 = \beta_0 = 0$. To learn this path, we solve the following Ordinary Differential Equation (ODE):

$$\frac{d}{dt} X_t = u_t(X_t), \tag{2}$$

where $u_t$ is a time-dependent velocity field, also referred to as the *drift*, typically parameterized by a neural network $u_t^\theta$ which trained by minimizing:

$$\mathcal{L}_{FM}(\theta) = \mathbb{E}_{t, X_t \sim p_t} \left[ D \left( \dot{X}_t, u_t^\theta(X_t) \right) \right], \tag{3}$$

where $\dot{X}_t = \frac{d}{dt} X_t$ denotes time-derivative of $X_t$ and $D$ denotes general Bregman Divergences measuring the dissimilarity. Once trained, this velocity field can be used to generate samples by integrating the ODE (Eq. 2) numerically. In our work, we solve the ODE by simplest method Euler integration:

$$X_{t+h} = X_t + h u_t^\theta(X_t), \tag{4}$$

where $h = \frac{1}{n}$ and $n$ is a hyperparameter representing the number of forward steps.

## 4 Method

### 4.1 ObjectNav Definition

We consider the ObjectGoal Navigation (ObjectNav) task, where an embodied agent is initialized at a random location in an unseen indoor environment and is instructed to navigate to an instance of a user-specified object category (e.g., a *chair*). At each timestep $t$, the agent receives egocentric observations $I_t$ (i.e., RGB-D images) and its pose $\omega_t$, which includes its location and orientation. Based on this sensory and positional input, the agent selects an action $a_t \in \mathcal{A}$, where $\mathcal{A}$ includes `move_forward`, `turn_left`, `turn_right`, and `stop`. The navigation episode terminates either when the agent issues the `stop` action or after a maximum of $T$ steps. An episode is deemed successful if the agent issues `stop` with the target object visible and within a threshold distance.

## 4.2 Navigation Overview

In line with previous works [9, 40, 72, 73, 74, 75], we incrementally build a local semantic map, but via scene segmentation instead of single-frame understanding. Our trained generative flow model then completes the partial map by generating the full semantic distribution, especially for unobserved areas. This distribution guides the agent to likely goal object locations, followed by deterministic local navigation. An overview of the navigation pipeline is illustrated in Fig. 2 (a).

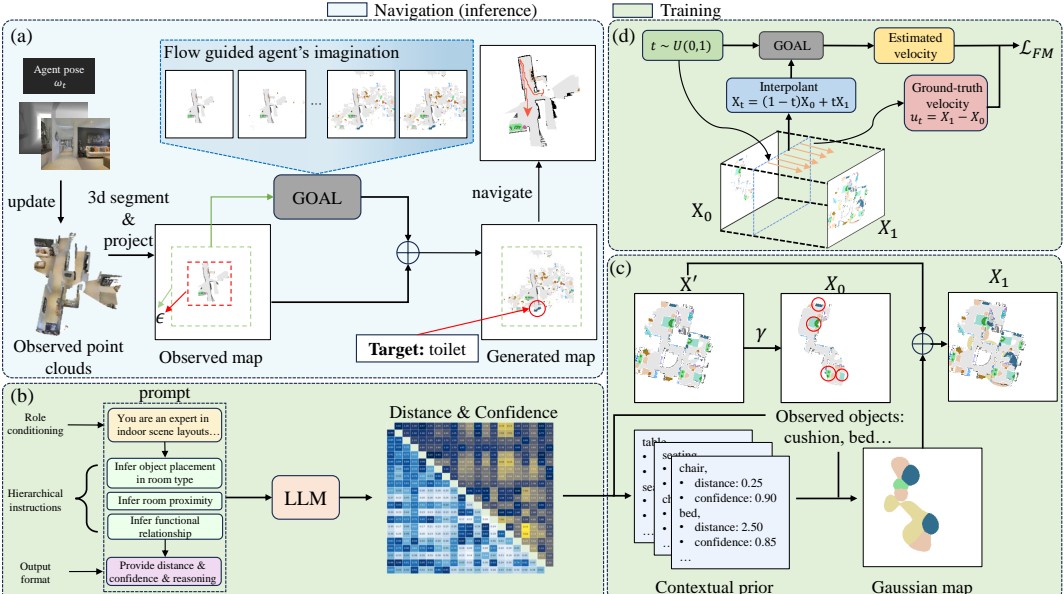

Figure 2: Overview of GOAL framework, with navigation (inference) in blue and training in green. (a) shows the navigation pipeline where the agent imagines future maps using a flow-guided model. (b) illustrates how we prompt a LLM with hierarchical instructions to generate contextual priors (for full prompt and response see Appendix E). (c) visualizes how we use LLM priors to construct data-dependent couplings. (d) demonstrates how the flow model is trained using these couplings through interpolated velocity supervision.

## 4.3 Generative Flow Models For Agent's Imagination

For training GOAL, we first prompt an LLM to obtain contextual priors between objects (Fig.2(b)), use these priors to construct dependent couplings as training samples (Fig.2(c)), and then train the model using standard interpolation scheme (Fig. 2(d)). We detail each step below.

**Prompting LLM for contextual prior.** Due to the scarcity of large-scale, densely annotated indoor scene datasets, models often struggle to generalize beyond seen environments. This is largely attributable to the limited diversity of contextual object arrangements in existing data. Here, we describe how we solve the problem by enriching training signals with commonsense knowledge about object co-occurrence, extracted from LLMs.

Specifically, we prompt LLM using various modern prompting techniques such as Chain-of-Thought (CoT) [58], Role Conditioning [44], Few-shot prompting [5], to generate the common distances between different objects $\mathcal{D} = \{d_{ij}\}_{i,j=1}^{N_c}$ and its confidences on each response $\mathcal{C} = \{c_{ij}\}_{i,j=1}^{N_c}$ (Fig. 2(b)). Then, given the partially observed semantic maps, we cluster the connected grids with the same semantic labels into objects $\mathcal{O} = \{o_i\}_{i=1}^{N}$, where $N$ is the number of observed objects. For each observed object $o_i$, we refer to the LLM responses for its likely co-occurring objects using preset distance threshold $\tau_d$ and confidence threshold $\tau_c$. Other objects $o_j$ that either has a distance to the observed object $d_{ij}$ within $\tau_d$ or a confidence score $c_{ij}$ larger than $\tau_c$ will be considered as co-occurring candidates. Formally, co-occurring candidates set for object $o_i$ is defined as:

$$\mathcal{N}(o_i) = \{o_j \mid (d_{ij} \leq \tau_d) \vee (c_{ij} \geq \tau_c)\}. \tag{5}$$

During training, every clustered observed object will randomly choose co-occurring object among their candidates set $\mathcal{N}(o_i)$ to enrich semantic context and avoid model collapse (In practice, rather

than strictly random selection, we actually prioritize the objects in the intersection of candidates set of multiple observed objects, to take account for a functional cluster with more than 2 objects.). Intuitively, if an object is expected to co-occur near an observed object, but has not yet been observed, it is likely located in the unobserved region of the scene. Thus we connect the clustered objects' centers with their nearest frontiers (the spatial boundaries between known free space and unexplored regions, by definition), and the centroid of the co-occurring objects will be on this connected line, offset by a distance $d_{ij}$. Specifically:

$$\boldsymbol{\mu_j} = \boldsymbol{\mu_i} + d_{ij}\boldsymbol{v}, \tag{6}$$

where $\boldsymbol{\mu_i}$ and $\boldsymbol{\mu_j}$ are the centroid of observed object $o_i$ and predicted centroid of co-occurring object $o_j$, and $\boldsymbol{v}$ is a unit vector pointing from $o_i$ to its nearest frontier grid. Then the confidence score $c_{ij}$ is converted to a standard deviation via linear transformation:

$$\sigma_{ij} = \sigma_{\min}c_{ij} + \sigma_{\max}(1 - c_{ij}), \tag{7}$$

where $\sigma_{\min}$ and $\sigma_{\max}$ are hyperparameters representing the lower and upper bounds of standard deviation, respectively. With the computed deviation and centroid, we model the co-occurring objects as two-dimensional Gaussian distributions, which are added across all observed objects:

$$p_{\text{LLM}}^{(j)} = \sum_{o_i \in \mathcal{O}} \frac{1}{2\pi\sigma_{ij}^{(1)}\sigma_{ij}^{(2)}} \exp\left\{-\frac{1}{2}\left[\left(\frac{x - \boldsymbol{\mu_j}^{(1)}}{\sigma_{ij}^{(1)}}\right)^2 + \left(\frac{y - \boldsymbol{\mu_j}^{(2)}}{\sigma_{ij}^{(2)}}\right)^2\right]\right\}, \; o_j \sim U(\mathcal{N}(o_i)), \tag{8}$$

where we take $\sigma_{ij}^{(1)} = \sigma_{ij}^{(2)} = \sigma_{ij}$, resulting in an isotropic Gaussian over the semantic map. Notation $U(X)$ refers to uniform distribution, which is used for random sampling. $p_{\text{LLM}}^{(j)}$ is then the LLM prior distribution reflecting the commonsense spatial expectations of object $o_j$ derived from LLMs. The final LLM prior is stacked across all channels:

$$p_{\text{LLM}} = \left[p_{\text{LLM}}^{(1)}, p_{\text{LLM}}^{(2)}, \ldots, p_{\text{LLM}}^{(N_c)}\right], \tag{9}$$

which has the same shape as the input semantic maps and is used to guide the flow model toward plausible object arrangements in unobserved regions of the scene during training.

**Building data-dependent couplings with LLM-derived supervision.** Following standard practice in diffusion and score-based models [21, 53], the source distribution is typically set to a standard Gaussian $\mathcal{N}(0, I)$, resulting in an independent coupling $\pi_{0,1}(X_0, X_1) = p(X_0)q(X_1)$. However, we find this strategy will complicate the model architecture with additional conditioning mechanism. Instead, we directly couple the partial semantic map with the LLM-enhanced target, eliminating the need for conditioning mechanisms like cross-attention and yielding more consistent and effective generation for navigation. Below, we describe how this dependent coupling is constructed.

During training, we have access to the full-scene semantic map $X'$. Following [40], we employ the Fast Marching Method (FMM) [49] to simulate a realistic navigation trajectory by planning a path between two randomly sampled points on the map. The visible region along this path serves as a binary mask $\gamma$, representing the observed area. To incorporate stochasticity and avoid model collapse, we still inject Gaussian noise with relatively small deviation $\Delta\sigma$ into the unobserved regions of the scene. Specifically, the source semantic map is defined as:

$$X_0 = \gamma \odot X' + \overline{\gamma} \odot \mathcal{N}(0, \Delta\sigma^2), \tag{10}$$

where $\odot$ denotes the Hadamard product, and $\overline{\gamma} = 1 - \gamma$ denotes the complement of the visibility mask. The target semantic map $X_1$ is constructed by adding LLM-derived priors to the unobserved regions of the full ground-truth map:

$$X_1 = \lambda\,\overline{\gamma} \odot p_{\text{LLM}} + X', \tag{11}$$

where $p_{\text{LLM}}$ is derived from Eq. 8 and Eq. 9, and $\lambda$ is a hyperparameter to control the prior strength. Notably, this formulation results in a data-dependent coupling between $X_0$ and $X_1$, since both maps are conditioned on the same ground-truth map $X'$ and share the same visibility mask $\gamma$. As a result, the joint distribution $\pi_{0,1}(X_0, X_1)$ described in Sec. 3 is no longer factorizable into independent marginals $p(X_0)q(X_1)$, but instead captures a strong, structured relationship between source and target data.

**Training.** Given the constructed data-dependent couplings, we adopt the Optimal Transport (OT) displacement interpolant as our interpolation scheme (see Eq. 1), defined as:

$$X_t = (1 - t)X_0 + tX_1. \tag{12}$$

For the Bregman Divergence term in Eq. 3, we use the Euclidean distance, resulting in the training objective being a Mean Squared Error (MSE) loss:

$$\mathcal{L}_{FM} = \mathbb{E}_{t, X_t \sim p_t} \left[ ||\dot{X}_t - u_\theta(X_t, t)||_2^2 \right], \tag{13}$$

where the ground-truth velocity field is given by $\dot{X}_t = X_1 - X_0$, derived from the linear interpolation in Eq. 12. For details of the training pipeline, refer to the Appendix B for pseudocode.

## 4.4 ObjectNav with GOAL

In this subsection, we detail the process of building and preprocessing the semantic map to serve as input to GOAL. followed by how the agent uses the output of it to guide exploration and take effective actions.

**Semantic map construction via 3D scene understanding.** We construct a semantic map by transforming RGB-D observations into 3D point clouds, segmenting them, and projecting the results to a top-down view, as detailed below. As described in Sec. 4.1, the agent receives RGB-D observations $I_t$ and its pose $\omega_t$ at each timestep $t$, along with the camera intrinsic matrix $P$ inherently available. The observations $I_t$ are first back-projected to point clouds by aligning the RGB values and depth values of each pixel. Then egocentric-to-geocentric transformation is conducted using $\omega_t$ and $P$. The registered point clouds will be segmented at once to take account of historical context, geometric structure, and achieve scene-level perception. Though modern architectures such as the Point Transformer series [61, 62, 76] have demonstrated strong performance in 3D scene understanding, we adopt a Sparse Convolutional network [18], the 3D counterpart of standard 2D CNN-based segmentation models, to reduce potential confounding effects from using highly expressive architectures. After segmentation, points are then projected to bird-eye view to form a local map $M_t \in \mathbb{R}^{(N_c+2) \times h \times w}$, where $N_c$ represents the number of semantic categories, and the additional 2 dimensions correspond to obstacles and free space, respectively (we denotes $M'_t \in \mathbb{R}^{N_c \times h \times w}$ to 'semantic map').

**Agent's imagination With GOAL.** To guide navigation toward the goal, the agent uses a generative model to imagine unobserved regions of the scene and select the grid with the highest probability to be the target as long-term way-point. Given the semantic map $M'_t$ at time $t$, we first compute the minimum bounding box of the observed area and crop a surrounding region scaled by a factor $\epsilon$. This factor controls the imagination's spatial extent: too small may limit foresight, while too large may reduce reliability. The cropped region is resized to a fixed shape $m'_t \in \mathbb{R}^{n \times L \times L}$ (with $L = 256$), where unobserved areas are injected with Gaussian noise similar to Eq.10. This map serves as the source sample $X_0$ in Eq.12 and is passed to the generative flow model, which applies iterative Euler steps (Eq. 4) to generate a complete semantic map. The output is then resized and merged back into the original map, resulting in the imagined full map $\hat{M}_t$. Finally, the grid cell with the highest value in the target object channel $c_g$ is selected as the long-term waypoint $g_t$:

$$g_t = \arg\max_{(h,w)} \hat{M}_{t(h,w)}^{(c_g)}. \tag{14}$$

**Navigation policy.** After determining the waypoint $g_t = (x_t, y_t)$, we follow prior works [9, 40, 75] by implementing a local planner based on the Fast Marching Method (FMM) [49], which computes the shortest path from the agent's current position to the waypoint using the occupancy channels of the semantic map. Unlike previous approaches that select waypoints at fixed intervals, we adaptively sample a new waypoint only when the agent is either too close to or too far from the previous one. This strategy reduces computational overhead while maintaining effective path planning.

## 5 Experiments

### 5.1 Experimental Setup

**Datasets.** We evaluate our method, **GOAL**, on the validation sets of two standard ObjectNav benchmarks: Gibson [63], Matterport3D (MP3D) [8]. Additionally, we perform transfer experiments

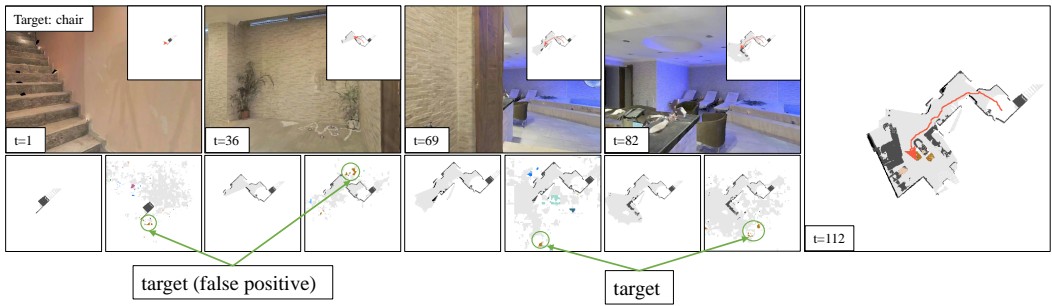

Figure 3: Visualization of navigation with GOAL on MP3D (val). The top row shows RGB observations and agent trajectories; the bottom row displays the observed semantic maps and generated full-scene maps.

Table 1: Transfer experiments results on HM3D. We compare the SR, SPL, and DTS of state-of-the-art methods in different settings and training set. * denotes our implementation using model weights and codes from official repositories, with improvement using scene segmentation.

| Method | Train set | LLM usage | | HM3D | | |
|---|---|---|---|---|---|---|
| | | Training | Inference | SR ↑ | SPL ↑ | DTS ↓ |
| ZSON [34] | HM3D | ✗ | ✗ | 25.5 | 12.6 | – |
| PixNav [6] | HM3D | ✗ | ✓ | 37.9 | 20.5 | – |
| ESC [77] | – | – | ✓ | 39.2 | 22.3 | – |
| VoroNav [60] | – | – | ✓ | 42.0 | 26.0 | – |
| PONI* [40] | MP3D | ✗ | ✗ | 41.8 | 20.1 | 4.63 |
| GOAL w/o LLM | MP3D | ✗ | ✗ | 47.6 | 22.5 | 4.14 |
| GOAL | MP3D | ✓ | ✗ | **48.8** | 23.1 | **4.11** |

by training on MP3D and evaluating on HM3D. For Gibson, we follow the tiny-split protocol from [9, 40], using 25 training and 5 validation scenes, with 1,000 validation episodes covering 6 target object categories. For MP3D, we use the standard Habitat simulator setting [35, 39, 55], which includes 56 training and 11 validation scenes, 2,195 validation episodes, and 21 target categories. For HM3D, we use only the 20 validation scenes, with 6 goal categories and 2,000 validation episodes for transfer evaluation. For details of target object categories, please refer to Appendix C.2.

**Evaluation metrics.** We adopt three standard metrics for evaluating the navigation performance. **SR** (Success Rate) indicates the proportion of success episodes. **SPL** (Success weighted by Path Length) represents the success rate of episodes weighted by path length, measuring the efficiency of navigation. **DTS** (Distance To Goal) is the distance to the goal at the end of the episode. For mathematical expression of these metrics, please refer to Appendix C.1.

**Implementation details.** For training of GOAL, we sample 400K sub-maps for training on each dataset. We implement GOAL based on DiT [37] models. We use AdamW [24] optimizer with a base learning rate of 1.5e-4, warmed up for 2 epochs, and applied cosine decay after that. The model is trained for 25 epochs, and exponential moving average (EMA) is used with a decay of 0.999 during training. We trained the GOAL model on 4 NVIDIA RTX 4090 GPUs with a batch size of 64 per GPU. For details of training scene segmentation module, please refer to Appendix.C.5.

## 5.2  Evaluation Results

**Visualization of navigation with GOAL.** Figure 3 illustrates an example episode where the agent is tasked with finding a chair. Initially, the agent is misled by a false positive prediction due to limited observations. As the agent explores and uncovers more of the environment, the model refines its prediction, effectively guiding exploration and ultimately leading agent to correctly identify and navigate to the target (rightmost column). Additional visualizations demonstrating generation quality and diversity are provided in Appendix F.

Table 2: Comparison between using dependent and independent couplings (Gaussian source). Navigation performance is reported in MP3D and Inference time is tested using an NVIDIA RTX 4090 GPU. The external knowledge is distilled from ChatGLM.

| Base PDF | Number of Parameters | Memory Usage | GFLOPs | Inference time | Navigation (MP3D) | | |
|---|---|---|---|---|---|---|---|
| | | | | | SR ↑ | SPL ↑ | DTS ↓ |
| $x_0 \sim \mathcal{N}(0, I)$ | 149.92M | 693.86 MB | 29.34 | 17.13 ms | 39.0 | 14.0 | 5.16 |
| $x_0 \sim \rho(X', \gamma)$ | **138.76M** | **562.57 MB** | **24.12** | **12.38 ms** | **41.5** | **15.5** | **4.85** |

Table 3: Ablation study of individual components on MP3D. 'LP' refers to distillation of LLM priors; 'SS' indicates scene segmentation.

| ID | Modules | | Navigation (MP3D) | | |
|---|---|---|---|---|---|
| | SS | LP | SR ↑ | SPL ↑ | DTS ↓ |
| 1 | | | 32.4 | 11.7 | 5.25 |
| 2 | ✓ | | 38.8 | 14.6 | 5.08 |
| 3 | ✓ | ✓ | **41.7** | **15.5** | **4.84** |

Table 4: Effectiveness of model variants. Navigation performance is tested in MP3D.

| Model Variants | LLM | Navigation (MP3D) | | |
|---|---|---|---|---|
| | | SR ↑ | SPL ↑ | DTS ↓ |
| DiT-B | ChatGLM | 41.5 | 15.5 | 4.85 |
| DiT-L | ChatGLM | 40.4 | 14.9 | 5.01 |
| DiT-B | Deepseek | 40.9 | 15.0 | 4.89 |
| DiT-L | Deepseek | 40.3 | 14.7 | 4.97 |
| DiT-B | ChatGPT | **41.7** | **15.5** | **4.84** |
| DiT-L | ChatGPT | 40.5 | 15.0 | 5.09 |

**Generalizability of GOAL.** We assess GOAL's generalization ability by training it on MP3D and directly evaluating on the HM3D dataset. We compare its performance against state-of-the-art methods across various training settings. As shown in Tab. 1, GOAL significantly outperforms prior methods, including those that heavily rely on LLMs or are trained directly on HM3D, demonstrating strong generalization capabilities. Notably, even the generative flow model without LLM supervision achieves competitive transfer performance, highlighting the benefits of generative modeling and diverse generation.

**Effectiveness of data-dependent couplings.** We compare bridging data-dependent couplings described in this paper and the traditional cross-attention method in DiT[37]. As shown in Tab. 2, building dependent couplings yields better navigation performance while simplifying model architecture and reducing inference time. Note that to condition the flow model on the partially observed semantic map, we introduce extra convolutional encoder to downsample the partial maps and then feed the patches to cross-attention layers of each DiT blocks, as an expressive mechanism to process and fuse the complex semantic map.

**Ablation study on LLM prior and scene segmentation components.** As shown in Tab. 3, scene segmentation alone significantly improves performance by enabling more consistent and complete scene understanding, already establishing a strong baseline (row 2). Notably, further integrating the LLM prior on top of this strong foundation yields a substantial additional gain (row 3), despite the typical difficulty of improving over high-performing baselines. This highlights the effectiveness of the LLM prior, and suggests that our bridging scheme between partial maps and full semantic distributions requires a reliable understanding of the observed scene.

**Effectiveness of Flow Matching.** Since GOAL shares the objective of inferring unobserved scene semantics with prior approaches [17, 25, 75], it is essential to compare the Flow Matching algorithm we adopt with these baseline prediction methods. As SGM [75] is the most recent approach in this line of work, we compare Flow Matching [26] with the masked autoencoder (MAE) [19] used in SGM, replacing our proposed scene segmentation module with the widely adopted RedNet [23] for a fair

Table 5: Comparison between Flow Matching (FM) and Masked Autoencoder (MAE) for semantics imagination. Models are trained on MP3D and evaluated on different datasets.

| Alg. | Eval. Dataset | SR ↑ | SPL ↑ | DTS ↓ |
|---|---|---|---|---|
| MAE | MP3D | 31.9 | **11.71** | 5.31 |
| FM | MP3D | **32.4** | 11.67 | **5.25** |
| MAE | HM3D | 32.1 | **14.7** | 4.85 |
| FM | HM3D | **35.9** | 14.69 | **4.65** |

Table 6: Object-goal navigation results on Gibson and MP3D. We compare the SR, SPL and DTS of state-of-the-art methods in different settings. For SemExp [9], L2M [17] and Stubborn [32], we report results from [74]. For SSCNav [25], we report results from [75]. '-' under *Training* indicates zero-shot methods that do not require any training. '_' means the second best results.

| Method | Venues | LLM usage | | Gibson | | | MP3D | | |
|---|---|---|---|---|---|---|---|---|---|
| | | Training | Inference | SR ↑ | SPL ↑ | DTS ↓ | SR ↑ | SPL ↑ | DTS ↓ |
| Semexp [9] | NeurIPS 20 | ✗ | ✗ | 71.1 | 39.6 | 1.39 | 28.3 | 10.9 | 6.06 |
| SSC-Nav [25] | ICRA 21 | ✗ | ✗ | – | – | – | 27.1 | 11.2 | 5.71 |
| PONI [40] | CVPR 22 | ✗ | ✗ | 73.6 | 41.0 | 1.25 | 31.8 | 12.1 | 5.10 |
| L2M [17] | ICLR 22 | ✗ | ✗ | – | – | – | 32.1 | 11.0 | 5.12 |
| Stubborn [32] | IROS 22 | ✗ | ✗ | – | – | – | 31.2 | 13.5 | 5.01 |
| CoW [16] | CVPR 23 | – | ✗ | – | – | – | 7.4 | 3.7 | – |
| 3DAware [74] | CVPR 23 | ✗ | ✗ | 74.5 | 42.1 | 1.16 | 34.0 | 14.6 | **4.78** |
| T-Diff [72] | NeurIPS 24 | ✗ | ✗ | 79.6 | **44.9** | 1.00 | 39.6 | 15.2 | 5.16 |
| L3MVN [71] | IROS 23 | – | ✓ | 76.1 | 37.7 | 1.10 | 34.9 | 14.5 | – |
| SG-Nav [69] | NeurIPS 24 | – | ✓ | – | – | – | 40.2 | 16.0 | – |
| UniGoal [70] | CVPR 25 | – | ✓ | – | – | – | 41.0 | **16.4** | – |
| SGM [75] | CVPR 24 | ✓ | ✓ | 78.0 | 44.0 | 1.11 | 37.7 | 14.7 | 4.93 |
| GOAL | Proposed | ✓ | ✗ | **83.5** | 44.2 | **0.83** | **41.7** | 15.5 | 4.84 |

comparison. As shown in Table 5, while MAE and Flow Matching achieve comparable performance on MP3D, Flow Matching demonstrates stronger generalization in the transfer setting, where models are trained on MP3D and evaluated on HM3D.

**Effectiveness of model variants and LLMs.** As shown in Tab.4, increasing model complexity (e.g., using DiT-L) yields little to no performance gain, and may even degrade performance compared to the base model (DiT-B). We hypothesize this is due to overfitting, as complex models tend to overfit when trained on limited and less diverse data, even in a generative setting. To assess the impact of LLMs in GOAL, we compare three models: ChatGLM-4-plus [14], DeepSeek-R1 [12], and GPT-4 [36]. As shown in Tab. 4, the overall navigation performance remains similar across LLMs (though we observe significant per-scene variance). Since a number of prior works [69, 75, 77] also report minimal differences between LLM variants, we hypothesize that current evaluation datasets are too small and biased to reliably reflect the effects of LLM choice.

**Comparison with related works.** We evaluate the performance of our method, GOAL, on the ObjectNav task by comparing it with relevant baselines, categorized by their use of LLMs during training and inference. SemExp [9] first introduced semantic reasoning into object-goal navigation. PONI [40] improves it using supervised learning to predict two potential functions, avoiding the inefficiencies of reinforcement learning. L2M [17] and SSC-Nav [25] improve navigation by predicting the full top-down semantic map. SGM [75] further advances these approaches using the MAE algorithm [19] to train a ViT [13] model in a self-supervised manner for full-scene semantic imagination. To enhance generation capabilities, SGM also prompts an LLM for contextual object information, but requires it during both training and inference, introducing additional complexity via a cross-attention mechanism. In contrast, methods like L3MVN [71], SG-Nav [69], and UniGoal [70] rely solely on LLMs to achieve zero-shot performance. While these approaches reduce training requirements, they often suffer from drawbacks such as API latency and excessive memory consumption.

As shown in Tab. 6. GOAL consistently outperforms the current state-of-the-art approach [75], across all metrics and datasets. Although the improvement in SPL is relatively small, this can be attributed to the modified local policy we employ, described in Sec. 4.4. Specifically, the agent only updates its long-term goal when it becomes significantly closer to or farther from the previous one. As a result, the agent's ability to correct false positive predictions is limited, even when new observations provide better guidance, leading to relatively long path length. And we stress that this limitation can be solved by applying modern techniques [15, 29, 33, 51, 52] to reach faster inference for generative flow and adopt the general policy of changing long-term goal at fixed interval. We leave this for future work as these techniques will introduce complex formulation.

# 6 Discussions and Limitations

There are several technical limitations and potential further improvements to consider.

First, we aim to introduce the generative flow matching algorithm into ObjectNav task, so our implementations adhers to the core theory in initial paper. However, recent advances in FM for natural image generation suggests some powerful training techniques to be used, such as time discretizations and time shift, as well as techniques that reduce the Number of Evaluations (NFEs) of flow matching[15, 29, 33, 51, 52]. Incorporating these techniques could enhance training stability and sampling efficiency, potentially yielding more robust navigation policies.

Second, our method operates on a fixed-dimensional semantic map, with channels corresponding to a predefined set of object categories. This design inherently restricts generalization to open-vocabulary or zero-shot settings, where novel object classes may appear at test time.

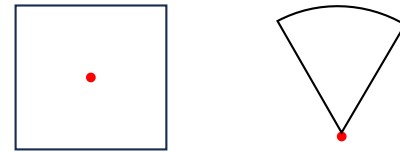

Figure 4: Comparison between the simulated visible area and actual visible area given the agent position (red dot). The left shows the simulated mask adopted by [40] and our work, while the right shows the actual mask, revealing a substantial gap.

Finally, we follow the setting of [40] by computing a path between two randomly selected points on the ground-truth semantic map, where a rectangular region centered at points along the path is treated as the visible area to generate the partially observed maps for training GOAL. However, we found a significant gap between these simulated partial observations and those encountered during real navigation. In practice, the agent's visible area is fan-shaped rather than rectangular (See Fig. 4). Designing training samples that better simulate actual agent observations could substantially improve model performance.

# 7 Conclusion

In this work, we propose GOAL, a generative flow model that distills rich contextual priors from LLM into its training supervision. We show that data-dependent couplings between partially observed maps and full semantic distributions significantly improve generation quality, outperforming traditional independent couplings commonly used in natural image domains. Additionally, we introduce a scene segmentation module to enhance holistic, geometry-aware, and temporally consistent scene understanding. Experimental results on large-scale datasets Gibson and MP3D validate the effectiveness of GOAL, while cross-dataset transfer experiments on HM3D further highlight its strong generalization capabilities.

# 8 Acknowledgments

This work was supported partially by NSFC(U21A20471,92470202, 62206315), by the National Key Research and Development Program of China (2023YFA1008503), Guangdong NSF Project (No. 2023B1515040025, No. 2024A1515010101), Guangzhou Basic and Applied Basic Research Scheme (No. 2024A04J4067).

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

# Appendix

We provide additional information about our method and experiments in the appendix. Below is a summary of the sections:

- Appendix A presents the strict mathematical formulation of Conditional Flow Matching (CFM), with additional notation omitted in the main text for simplicity.
- Appendix B describes the training algorithm with pseudocodes.
- Appendix C outlines the experimental setup and further implementation details.
- Appendix D provides more experimental results and analysis.
- Appendix E includes the prompts used for querying the LLM and its responses.
- Appendix F provides additional visualizations of our results.

## Appendix A    Strict Formulation of CFM

The Flow Matching loss defined in Equation 3 is, in practice, not directly solvable (since the target velocity $\dot{X}_t$ is not tractable). Throughout the paper, we refer to the Conditional Flow Matching (CFM) algorithm without explicitly including the conditioning variables in the notation, for simplicity. In this section, we present the strict and complete formulation of the Conditional Flow Matching algorithm.

Conditioning design in CFM vary, examples include conditioning on source sample $X_0$, the target sample $X_1$, or joint coupling $(X_0, X_1)$, and they are essentially equivalent. We exhibit the general formulation (conditioned on any random variable $Z$).

Following [57], suppose that *marginal probability path* $p_t(x)$ is a mixture of probability paths $p_t(x|z)$ that vary with some conditioning variable $z$:

$$p_t(x) = \int p_t(x \mid z)q(z)dz. \tag{15}$$

The *marginal velocity field*, which generates this marginal probability path, is given by averaging the conditional velocity field $u_t(x \mid z)$ across the condition $z$:

$$u_t(x) = \int u_t(x \mid z)p_{Z|t}(z \mid x)dz = \mathbb{E}\left[u_t(X_t|Z) \mid X_t = x\right] \tag{16}$$

Conditional Flow Matching Loss is then defined to be:

$$\mathcal{L}_{CFM}(\theta) = \mathbb{E}_{t,Z,X_t \sim p_{t|Z}(\cdot|Z)}\left[D\left(u_t(X_t \mid Z), u_t^\theta(X_t)\right)\right]. \tag{17}$$

In practice, we actually apply data-coupling conditioning (namely $Z = (X_0, X_1)$) as shown in Eq. 12, $X_t \sim p_t$ is given by a linear combination of $X_0$ and $X_1$, and velocity field is hence given by $X_1 - X_0$. Then the loss in Eq.13 can indeed solve the Flow matching problem introduced in Sec. 3.

## Appendix B    Algorithm

Algorithm 1 outlines the procedure for constructing data-dependent couplings and training the GOAL model. The corresponding sampling process is detailed in Algorithm 2. We stress that the sampling requires no additional pre-processing, making it a plug-and-play module.

## Appendix C    Experimental setup and implementation details

### C.1    Metrics

As mentioned in the main text, we select **SR**, **SPL**, **DTS** as the evaluation metrics for ObjectNav performance. Following we give their mathematical formulation and explaination.

---
**Algorithm 1** Training algorithm for **GOAL**
---
1: **Input:** dataset $\mathcal{X}$, initial model parameter $\theta$, learning rate $\eta$, distance matrix $\mathcal{D}$ and confidence matrix $\mathcal{C}$ from LLM responses.
2: **repeat**
3:  sample $X' \sim \mathcal{X}$
4:  Randomly sample two points $g_1, g_2$ on the grid
5:  Compute visible mask $\gamma$ by planning a path from $g_1$ to $g_2$
6:  $X_0 \leftarrow \gamma \odot X'$
7:  Cluster $X_0$ into observed objects $\{o_i\}_{i=1}^N$
8:  Initialize $p_{\text{LLM}}$ as a zero vector with the same shape as $X'$
9:  **for** each $o_i$ in $\{o_i\}_{i=1}^N$ **do**
10:   $p_{\text{LLM}}^{(j)} \leftarrow p_{\text{LLM}}^{(j)} + \texttt{ComputeLLMPrior}(o_i, \mathcal{D}, \mathcal{C})$     $\triangleright$ See Eq. 8
11:   $X_1^{(j)} \leftarrow X_1^{(j)} + \lambda \overline{\gamma} \odot p_{\text{LLM}}^{(j)}$
12:  **end for**
13:  $X_0 \leftarrow X_0 + \overline{\gamma} \odot \mathcal{N}(0, \Delta\sigma^2)$
14:  $t \sim \mathcal{U}[0, 1]$
15:  $X_t \leftarrow (1-t)X_0 + tX_1$
16:  $\hat{u}_t \leftarrow u_\theta(X_t, t)$
17:  $\mathcal{L} \leftarrow \text{MSE}(\hat{u}_t, X_1 - X_0)$
18:  $\theta \leftarrow \theta - \eta\nabla_\theta\mathcal{L}$
19: **until** convergence
---

---
**Algorithm 2** Sampling algorithm for **GOAL**
---
1: **Input:** trained GOAL model $u_\theta$, partially observed semantic map $M$, number of steps $n$, standard deviation $\Delta\sigma$.
2: $\gamma \leftarrow$ mask of empty area of $M$
3: $M \leftarrow M + \overline{\gamma} \odot \mathcal{N}(0, \Delta\sigma^2)$
4: **for** $k \leftarrow 1$ to $n$ **do**
5:  $t_k \leftarrow \frac{k}{n}$
6:  $\Delta M \leftarrow u_\theta(M, t_k)$
7:  $M \leftarrow M + \frac{1}{n}\Delta M$
8: **end for**
9: **return** $M$
---

**Success Rate** (**SR**) represents the agent's accuracy in reaching the user-specified object goal, where higher values indicates better performance:

$$SR = \frac{1}{N}\sum_{i=1}^N S_i, \tag{18}$$

where $N$ is the number of validation episodes and $S_i$ indicates whether the i-th episode is successful.

**Success weighted by Path Length** (**SPL**) evaluates success relative to the shortest path, normalized by the actual path length agent takes:

$$SPL = \frac{1}{N}\sum_{i=1}^N S_i \frac{l_i^*}{\max(l_i, l_i^*)}, \tag{19}$$

where $l_i^*$ denotes the shortest path length and $l_i$ is the actual path length agent takes.

**Distance To Goal** (**DTS**) measure the distance of agent towards the target object when the episode ends:

$$DTS = \frac{1}{N}\sum_{i=1}^N \max(L_{i,g} - \xi, 0), \tag{20}$$

where $L_{i,g}$ is the distance between agent and goal, and $\xi$ is the success threshold (0.1 meter in MP3D).

Table 7: Chosen object categories in Gibson [63], MP3D [8] and HM3D [41]

| Dataset | Training | Evaluating |
|---|---|---|
| Gibson | *chair, couch, potted plant, bed, toilet, dining-table, tv, oven, sink, refrigerator, book, clock, vase, cup, bottle* | *chair, couch, tv, bed, toilet, potted plant* |
| MP3D | *chair, table, picture, cabinet, cushion, sofa, bed, chest of drawers, plant, sink, toilet, stool, towel, tv monitor, shower, bathtub, counter, fireplace, gym equipment, seating, clothes* | *chair, table, picture, cabinet, cushion, sofa, bed, chest of drawers, plant, sink, toilet, stool, towel, tv monitor, shower, bathtub, counter, fireplace, gym equipment, seating, clothes* |
| HM3D | – | *chair, couch, potted plant, bed, toilet, tv* |

## C.2 Object categories

Following the setup of previous works [40, 72, 75], we adopt 15 categories for training and 6 categories for validation in the Gibson dataset, and 21 categories for both training and validation in the MP3D dataset. Additionally, validation episodes in the HM3D dataset contain 6 categories. The adopted categories are detailed in Table 7.

## C.3 Trivial hyper-parameters

There are a number of trivial hyper-parameters which are not tuned, we detail the choices we adopt intuitively in Tab. 8 for better reproduction.

| Hyper-Parameters | Values |
|---|---|
| $\tau_d$ (Eq. 5) | 2.5 |
| $\tau_c$ (Eq. 5) | 0.85 |
| $\sigma_{\min}$ (Eq. 7) | 20 |
| $\sigma_{\max}$ (Eq. 7) | 50 |
| $\lambda$ (Eq. 8) | 1500 |
| $\Delta\sigma$ (Eq. 8) | 0.01 |

Table 8: Values for trivial hyper-parameters

## C.4 Memory and Time Cost Analysis

Our method involves maintaining point cloud representations and performing scene segmentation, along with generative modeling for exploration. These components naturally raise concerns about memory and computation overhead.

Unlike semantic maps, whose memory usage is fixed by tensor grid dimensions, point cloud memory consumption varies significantly across scenes and episodes, depending on how much of the environment has been explored. As a result, a scene-independent comparison is difficult. In practice, we observed that running 6 parallel threads on a 24GB NVIDIA RTX 3090 consumes approximately 22GB of memory. For comparison, PONI [40] reports around 20GB under similar settings, indicating our approach adds roughly 350MB per thread.

Time cost also varies across scenes and depends on the agent's waypoint update frequency. Since GOAL is only invoked when the agent is either close to or far from the previous target, inference is sparse and input-dependent. Empirically, running 6 threads in parallel yields an average FPS between 1.2 and 1.8, while PONI ranges from 1.5 to 3.5. Despite this, we consider the trade-off worthwhile, as our method achieves over 30% improvement in success rate (SR), the primary metric of interest.

## C.5 Training of scene segmentation model

We train a Sparse UNet [18, 46] with the assistance of Pointcept [11] codebase, using a sum of weighted Cross-Entropy loss $\mathcal{L}_{CE}$ and Lovász-Softmax loss [3] $\mathcal{L}_{LZ}$ as training objectives:

$$\mathcal{L} = \mathcal{L}_{CE} + \mathcal{L}_{LZ}. \tag{21}$$

For optimizer, we simply select Stochastic gradient descent (SGD) [4] with a base learning rate of 0.05, momentum of 0.9 and a weight decay of 0.0001. Additionally, the first 5% of the training steps are used for warm-up, and the learning rate smoothly decays using cosine annealing over the remaining 95% of the training steps. We train the scene segmentation model on 2 NVIDIA RTX 4090 GPUs with a total batch size of 64.

# Appendix D  More experimental results

## D.1  Hyper-parameters tuning

We tune two key hyperparameters: the expansion ratio of the observed map $\epsilon$, and the number of Euler steps $n$ used during generation. The effect of varying the number of Euler steps $n$ is shown in Fig. 5. Navigation performance generally improves with more steps, saturating around $n = 96$. As discussed in Sec.5.2, while the overall performance across different LLMs is comparable, each exhibits a distinct preference for the expansion ratio $\epsilon$. Therefore, we tune $\epsilon$ individually for each LLM, as shown in Fig.6. We observe that the flow model distilled with external knowledge from ChatGPT produces more reliable semantic distributions with a larger expansion ratio.

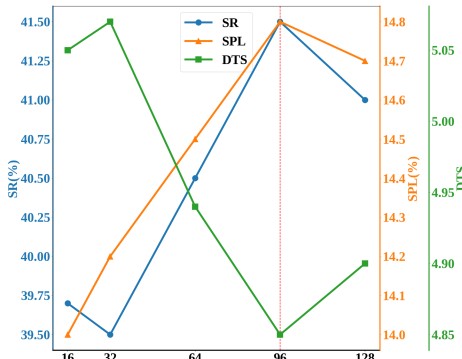

Figure 5: Effect of the number of Euler steps $n$ on navigation performance.

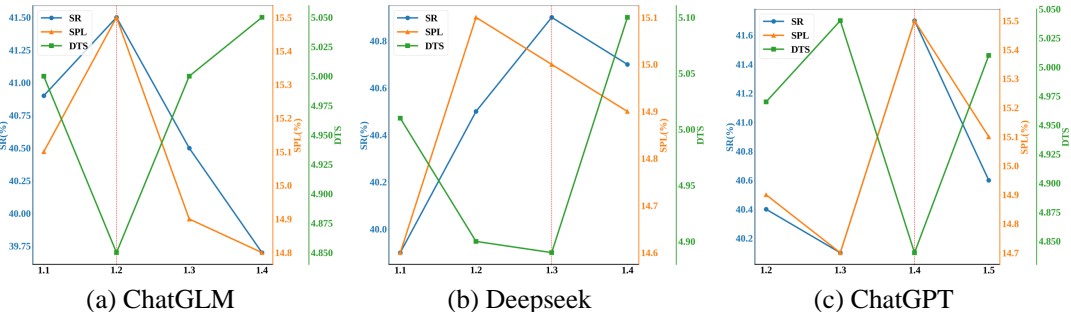

      (a) ChatGLM                     (b) Deepseek                  (c) ChatGPT

Figure 6: Tuning curve for hyper-parameter $\epsilon$ across different LLMs.

## D.2  Evaluation Variability and Error Analysis

While probabilistic generative schemes offer improved generalization, they naturally introduce concerns about evaluation stability due to inherent stochasticity. To assess the robustness of our model, we conduct additional evaluations on the MP3D dataset using different random seeds (42, 75, 100, 123, 3407). The resulting success rates are 41.0%, 41.6%, and 41.7%, 41.6%, 42.2% respectively, yielding an average success rate of $41.6\% \pm 0.4\%$. These results indicate that our flow model can reliably capture the semantic distribution, despite the stochasticity introduced by the generative process. We report the result with seed 100 in the main text, as all experiments and hyperparameter tuning were conducted under this setting.

# Appendix E  Prompts and LLM responses

## E.1  Prompts

We provide an example of prompts that query LLMs for objects contextual information. Specifically, we first condition the LLM with its role in the system prompt. Next, we provide contextual information, followed by a chain-of-thought [58] style of hierarchical prompting, which involves scenes, rooms, and objects. Moreover, for each step, we also provide some positive and negative examples to serve as few-shot learning samples. In addition to the distance-confidence pair, we further require the LLMs to output a brief reasoning for their responses. An example prompt is as follows:

---

**Prompt**

**System Prompt:** You are an expert in indoor scene layouts, with strong reasoning skills regarding object co-occurrence. Your task is to infer the typical distances between different object types in indoor environments, considering both object placement patterns and functional relationships. Output your answers in a clear, structured format with a confidence level that reflects the uncertainty of each estimate.

**User Prompt:** In indoor scenes, object layouts typically follow certain patterns; for example, chairs are usually placed around a table but are unlikely to be near a toilet. Suppose you are analyzing a large-scale indoor scene (e.g., a house with multiple rooms such as living rooms, bedrooms, bathrooms, etc.). Given the following list of objects: [chair, table, ...] and a specific *central object* in it, your tasks are as follows, step-by-step:

1. **Inferring Object Placement:** Determine where each object is typically placed in the scene based on common indoor layouts. For example:
   - Toilet → bathroom

2. **Room Proximity:** Identify the typical rooms surrounding the room where the *central object* is placed. For instance:
   - Chairs (living room) → nearby: kitchen, hallway
   - Bed (bedroom) → nearby: bathroom, closet

3. **Functional Relationships:** Consider the functional relationships between the *central object* and other objects:
   - Chair & table: strong (dining/study areas)
   - Chair & toilet: weak (different functional zones)

4. **Distance Inference:** Based on the inferred scene layout and functional relationships, estimate the typical distance between the central object and other objects. For example, if the *central object* is 'chair', then:
   - table to chair: close (functional pair)
   - Chair to closet: relatively farther away (possibly the same room but with weak functional relationship)
   - Chair to toilet: large distance (different functional zones)

5. **Provide Distance and Confidence:** Given *central object* = ..., for each object provide:
   - Distance: **continuous float value in meters**
   - Confidence: **score between 0 (uncertain) and 1 (certain)**
   - Reasoning: A brief explanation of how the distance and confidence were determined, considering the room layout and common object placements

---

### E.2 Responses

Few examples for LLMs response are as follows:

---

🤖 **Response**
*central object*: table
- *quried object*: chair
    - distance: **0.5**
    - confidence: **0.95**
    - reasoning:
      Chairs are commonly placed directly around a table, especially in dining areas or meeting rooms, making the distance very short and the placement highly predictable.

---

- *quried object*: picuture
  - distance: **1.5**
  - confidence: **0.8**
  - reasoning:
    Pictures are often hung on walls near tables, especially in living or dining rooms, but not as close as chairs, leading to a moderate distance with good confidence.

- *quried object*: counter
  - distance: **4.5**
  - confidence: **0.65**
  - reasoning:
    Counters are often found in kitchens, which may be adjacent to dining areas but not directly next to tables, leading to a moderate distance with moderate confidence.

- *quried object*: bathtub
  - distance: **8.0**
  - confidence: **0.5**
  - reasoning:
    Bathtubs are also found in bathrooms, separate from areas where tables are commonly placed, resulting in a large distance with low confidence.

  ......

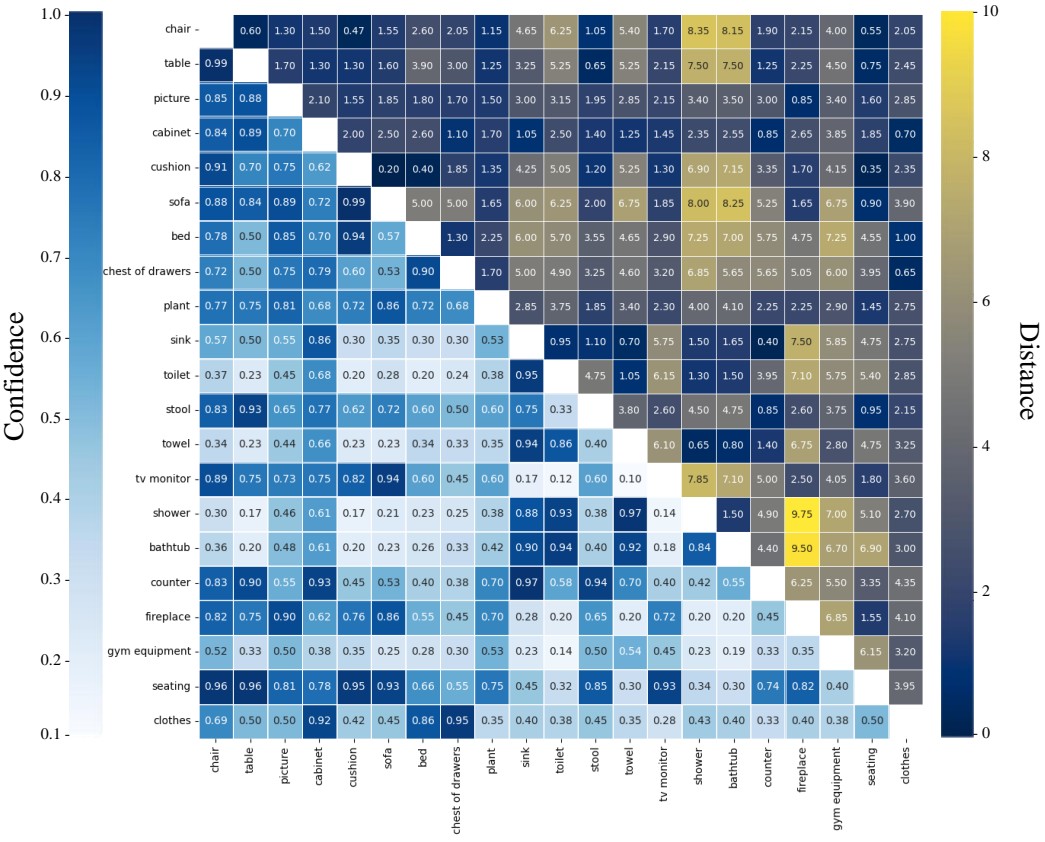

Figure 7: Structured distance matrix $\mathcal{D}$ and confidence matrix $\mathcal{C}$ obtained from GPT-4 for MP3D. We visualize the upper triangle of the distance matrix and the lower triangle of the confidence matrix within the same figure for compactness and clarity

Since the distance between objects should be bidirectional, we take the average of the symmetric distances and confidences, resulting in two symmetric matrices. Structured matrix representation of LLM response from GPT-4 is shown in Fig. 7.

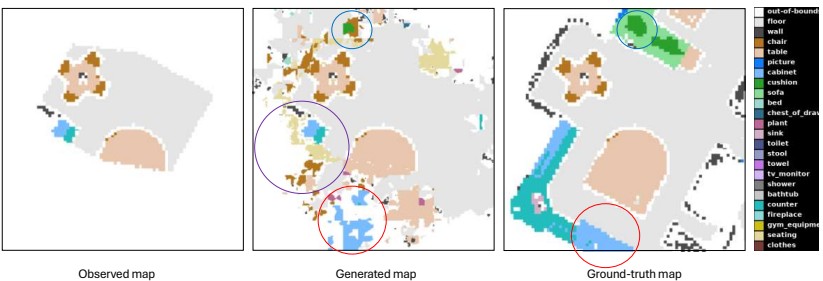

Figure 8: Example for semantic map generated by generative flow model. It successfully generates objects in GT semantic map like cabinet (highlighted in red circle) and cushion (highlighted in blue circle). Moreover, it also generate objects not in the ground-truth map but indeed reasonable, such as seating and chair (highlighted in purple circle) behind the table, which can improve the capability of generalization.

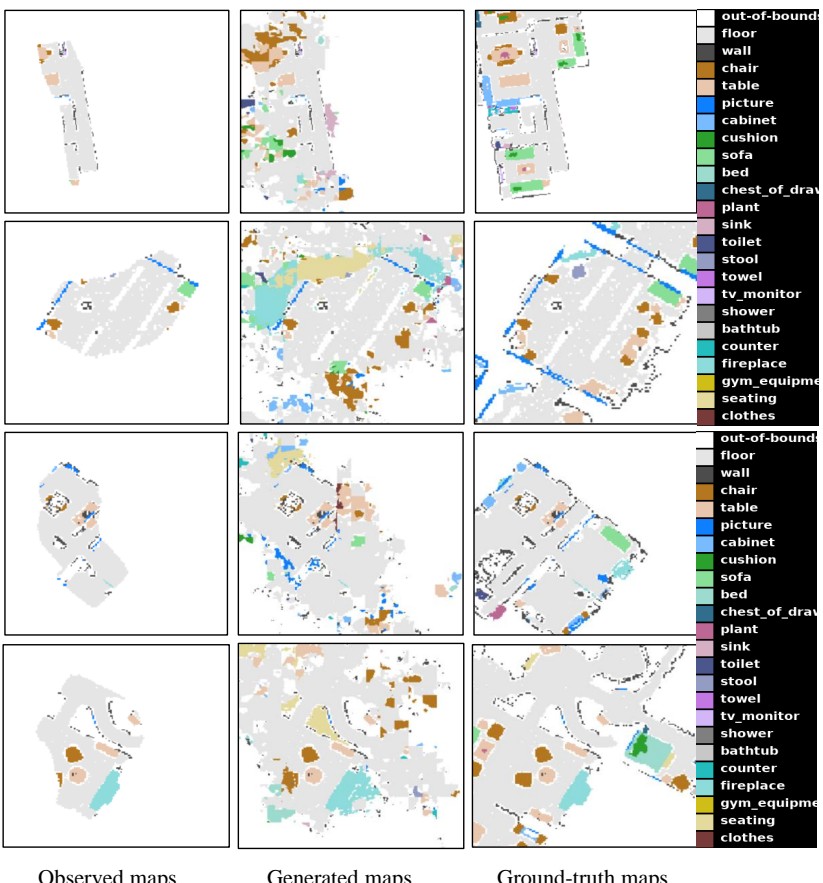

Figure 9: More visualizations for outputs of generative flow model.

# Appendix F   More visualizations and analysis

## F.1   visualizations for generative flow

In this subsection, we present visualizations and analyses of the outputs produced by our generative flow model. Generating a complete semantic distribution of a scene is an inherently challenging task and unlikely to be perfectly accurate. While the visualizations may not appear flawless, they significantly contribute to navigation performance. We begin with an illustrative example accompanied by detailed analysis (see Fig.8), followed by additional qualitative results (Fig.9). In this paper, we stress that indoor scene semantics can vary greatly, and multiple plausible distributions may exist given the same partial semantic map. To capture this diversity, GOAL adopts a probabilistic generation scheme, which enhances the model's ability to generalize to unseen environments. We showcase a sample of this generative diversity in Fig. 10. While we highlight examples where objects are generated near observed ones to provide intuitive evaluation, the diversity applies to the full semantic distribution and is not limited to individual object placements.

## F.2   visualizations for scene segmentation

In Fig. 11, we additionally present few visualizations for comparison between scene segmentation proposed in this paper and image segmentation traditionally adopted in ObjectNav.

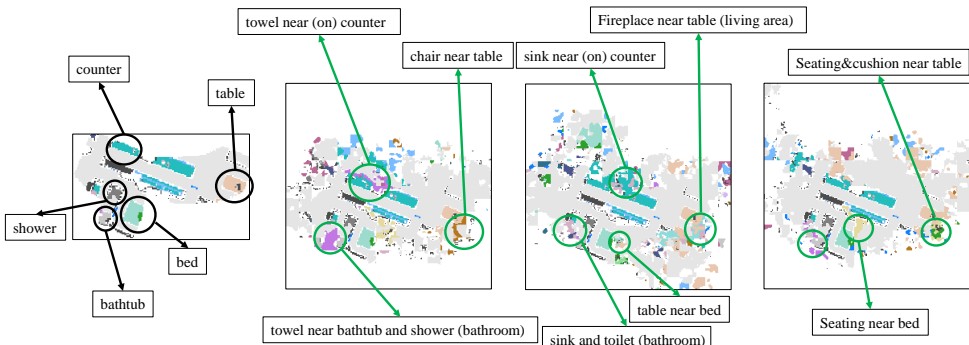

Figure 10: Visualization of generation diversity. Given a single partial map (left most), GOAL can generate multiple plausible full semantic distribution (right three).

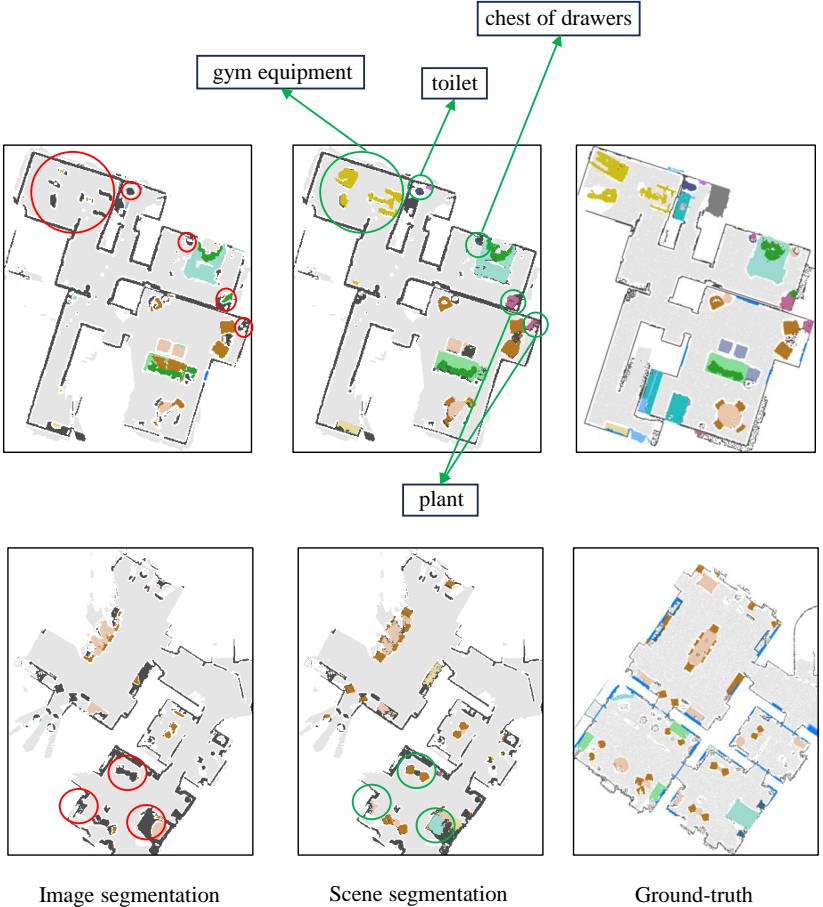

Figure 11: Comparison between built semantic maps using image segmentation models and scene segmentation models.

