# OpenReview forum: "Distilling LLM Prior to Flow Model for Generalizable Agent’s Imagination in Object Goal Navigation"
_NeurIPS.cc/2025/Conference — NeurIPS 2025 poster_

### Official Review · Reviewer_eNUZ · 2025-06-30

**Clarity:** 4
**Significance:** 3
**Originality:** 4
**Rating:** 5
**Confidence:** 5

**Summary:**

This paper proposes GOAL, a generative method for object-goal navigation. Previous methods build incremental BEV semantic map and apply discriminative method to infer the position to navigate. On the contrary, this paper model objectnav as a generative problem. It proposes a novel framework to distill LLM's common knowledge into a flow-based model, which guesses unseen region to facilitate navigation planning. Experimental results on several benchmarks demonstrates the effectiveness of GOAL.

**Questions:**

1. The choice of generation. Why generate semantic map rather than other representations like 3D scene graph? State-of-the-art methods like UniGoal leverages 3D scene graph rather than BEV semantic map.
2. Can this method be extended to more tasks rather than object goal?

**Ethical Concerns:**

["NO or VERY MINOR ethics concerns only"]

**Limitations:**

yes

**Quality:**

3

**Strengths And Weaknesses:**

# Strength:
1. Model map-based ObjNav as a generative problem is interesting and makes sense.
2. The technical contribution is sufficient.
3. During inference, LLM is not required, which may be friendly for on-device deployment.

# Weakness:
1. For 3D perception, the paper claims "To mitigate this, we aggregate past RGB-D observations into unified point clouds representations and perform joint segmentation using 3D perception models, inspired by how humans implicitly integrate multi-frame observations." However, it seems the authors only leverage per-frame 3D segmentation using sparseconv, without modeling the inter-frame temporal information. Applying online 3D perception may achieve better performance [1][2][3].
2. One advantage of GOAL is that it does not require LLM during inference. I'd like to know its inference speed compared with other methods. This should be demonstrated in experimental results.


[1] Fusion-aware point convolution for online semantic 3d scene segmentation. CVPR 2020.

[2] Memory-based adapters for online 3d scene perception. CVPR 2024.

[3] Embodiedsam: Online segment any 3d thing in real time. ICLR 2025.

---

> ### Author Rebuttal · Authors · 2025-07-31
>
> Thank you for your very positive and insightful review. We are delighted that you found our approach of modeling map-based ObjectNav as a generative problem interesting, our technical contributions sufficient, and our paper clear and well-illustrated. We also appreciate your recognition of the significant advantage of not requiring an LLM during inference for on-device deployment. We address your points below.
>
> # Weakness1: Suggestion on using online 3D perception method
>
> We thank the reviewer for this constructive advice and for highlighting the valuable advancements in online 3D perception methods ([1], [2], [3]). We completely agree that explicitly leveraging inter-frame temporal information, as these methods do, is highly suitable for tasks involving streaming RGB-D observations and holds significant potential to enhance the fidelity of our observed semantic maps.
>
> While our current approach implicitly integrates multi-frame data by aggregating past RGB-D observations into unified point clouds, integrating advanced online perception modules like those you referenced presents a **significant engineering challenge and requires substantial architectural changes** to our existing framework. **Thoroughly implementing and evaluating such a complex integration is unfortunately beyond the scope and feasibility of this rebuttal period**.
>
> However, we fully recognize its potential to further enhance our observed semantic map fidelity. We will ensure to discuss these relevant methods more extensively in the our revised paper, highlighting them as a valuable avenue for continued research.
>
> # Weakness2: Inference speed comparison
>
> **There is overwhelming difference in magnitude between methods prompting LLM online and ours.** Since UniGoal [4] has not open-sourced its code for object-goal navigation evaluation, we reimplemented SG-Nav [5], which represents a similar paradigm of real-time LLM interaction.
> |Method| average time per episode|
> |-----|-------|
> |SG-Nav|1h 54m|
> |GOAL | 2m 30s |
>
> **Note on Comparison Fairness**: It is important to acknowledge that this comparison is not perfectly balanced. We were only able to test SG-Nav on 6 episodes (which already took approximately 11.5 hours on our setup), whereas GOAL was tested on 220 episodes in parallel on two RTX 3090 GPUs, with the reported time being the average. Despite this slight unfairness in the number of episodes, the overwhelming difference in magnitude demonstrates the profound superiority of our method's inference speed. Our framework's design, which distills LLM knowledge offline, eliminates the latency of real-time LLM calls, resulting in an order-of-magnitude faster execution for entire navigation episodes. Reviewer can further refer to the issues of SG-Nav repo, where relevant questions on inference time can validate our results.
>
> We also want to emphasize that while SG-Nav incurs significant inference time, it offers valuable benefits from its training-free and open-world setting, which is a different and equally valuable research direction. Our work presents an alternative trade-off: leveraging offline LLM distillation for high efficiency during deployment.
>
> # Question1: The choice of generation.
>
> We thank the reviewer for their valuable question regarding the choice of semantic map representation over alternatives like 3D scene graphs. Our perspective is that while any representation can be adopted with specific technical designs (For example, leveraging discrete flow matching algorithms with graph neural network backbones and suitable LLM augmentation can also generate a 3D scene graph), our core contribution lies in the generative modeling methodology and LLM prior distillation.
>
> Our decision to generate a BEV (Bird's-Eye View) semantic map is **primarily for convenience and direct applicability to navigation**. Even state-of-the-art methods that represent entire scenes with 3D scene graphs, such as SG-Nav and UniGoal, still **fundamentally rely on a basic BEV map representing occupancy and free space for navigation planning (FMM)**. Directly building the semantic map upon this necessary BEV foundation is therefore a natural and efficient choice for our framework.
>
>
>
> # Question2: Extension of the method to more tasks
>
> Yes, our method is extensible beyond object-goal navigation. The core strength of our approach lies in capturing inter-object relationships and co-occurrence patterns to generate plausible semantic layouts for unobserved regions. This fundamental capability makes it applicable to various other scene-level tasks such as 3D Indoor Scene Generation: Our method's ability to "imagine" coherent object placements and semantic distributions can provide valuable guidance for 3D indoor scene generation, where multiple objects need to be generated in a contextually appropriate manner. If integrated with techniques for generating 3D scene graphs (as discussed in Question 1), this could further benefit complex scene generation tasks based on scene graph[6].
>
> However, it is important to note that these extensions are not trivial and would require specific design choices and adaptations. Our current model primarily focuses on generating object presence and coarse spatial relationships crucial for navigation, without explicitly modeling fine-grained object sizes or other necessary detailed assets for tasks like direct scene rendering. Therefore, direct application without modification would be impractical, necessitating further research and development tailored to each specific task.
>
> We would discuss more about the broader impact and application in the appendix in the final version of paper.
>
>
> We hope these comprehensive responses and additional analyses address all your concerns. We are confident that these clarifications further strengthen the paper.
>
> [1] Fusion-aware point convolution for online semantic 3d scene segmentation. CVPR 2020.
>
> [2] Memory-based adapters for online 3d scene perception. CVPR 2024.
>
> [3] Embodiedsam: Online segment any 3d thing in real time. ICLR 2025.
>
> [4] UniGoal: Navigate to Any Goal in Zero-shot! CVPR 2025.
>
> [5] SG-Nav: Online 3D Scene Graph Prompting for LLM-based Zero-shot Object Navigation. NeurIPS 2024.
>
> [6] Controllable 3D Outdoor Scene Generation via Scene Graphs. ICCV 2025.

---

### Official Review · Reviewer_ZGge · 2025-07-03

**Clarity:** 3
**Significance:** 3
**Originality:** 2
**Rating:** 4
**Confidence:** 4

**Summary:**

The submission proposes GOAL, a generative flow-based framework that incorporates external knowledge from large language models to enhance the agent's ability to imagine unobserved regions in object goal navigation tasks. The method achieves state-of-the-art performance on MP3D and Gibson datasets and demonstrates strong generalization capabilities in transfer settings to HM3D.

**Questions:**

- When using models that can run on consumer-grade GPU, such as LLaMA 7B, for training, what is the performance of this method?

- Since the generated content of LLM only includes distance and confidence, is there a significant gap in the content generated by different LLMs? Please provide quantitative analysis or examples for illustration.

- For hyper-parameters tuning, is there a more reasonable interpretation of the current performance curve? Is it possible to propose a more systematic optimization approach?

- Table 2 takes the model's real-time performance into account, and dependent couplings achieve a significant reduction in inference time (from 17.13ms to 12.38ms). However, for an episode, it seems that map prediction for waypoint selection does not need to be performed at every step. So, how much delay will be caused in an entire episode?

**Ethical Concerns:**

["NO or VERY MINOR ethics concerns only"]

**Final Justification:**

My concerns have been partially addressed. I vote for borderline accept.

**Limitations:**

Yes

**Quality:**

3

**Strengths And Weaknesses:**

## Strength
- The method outperforms existing approaches on multiple benchmarks, including MP3D and Gibson.

- The submission provides experiments and ablation studies, demonstrating the effectiveness of each component of the framework and its overall robustness.

- The writing and illustration of the submission are convenient for understanding.

## Weakness
- The primary weakness of the submission is the lack of novelty in methods/architecture. Similar to SGM, it predicts unobserved maps to guide navigation, differing mainly in specific technical approaches. But I also question the necessity of the proposed methods.

- I question the necessity of the proposed methods:
  - Regarding the "Distilling LLM Prior", the data shown in Table 1 also indicate that the improvement brought by LLM is very limited (SR:47.6→48.8). And when using LLMs, the prompts in this submission only provide central objects without specific information about other actual objects surrounding them, lacking environmental distribution information. This approach is insufficient for handling complex environments.
  - Regarding the "Flow Model", Table 2 shows that the dependent couplings method(SR:41.5) outperforms independent couplings (SR:39.0), but the improvement is marginal. Given that dependent couplings rely on extensive hyper-parameter tuning (D.1 Hyper-parameters tuning), I find it difficult to be convinced of the effectiveness of the proposed method.

- Demonstrated in "D.1 Hyper-parameters tuning" of the submission, the hype-parameters mainly relies on experimental testing and optimization. The performance fluctuates significantly with several hyper-parameters, and there is a lack of reasonable explanation, which deepens the concerns about the reliability of the method.

- Although the experiment tested LLMs from different manufacturers, they are all LLMs with a large number of parameters (for example, DeepSeekR1 has 671B parameters), lacking the ability test of small-parameter models. However, works such as SG-Nav, often achieve good results on models with relatively small parameters (such as LLaMA 7B in SG-Nav).

---

> ### Author Rebuttal · Authors · 2025-07-31
>
> We appreciate your critical questions and concerns, which we address comprehensively below.
> # Weakness1: Novelty of the paper.
>
> Map prediction is a fundamental direction of ObjectNav, and a series of works (L2M [1], SSC-Nav [2], SGM[3]) have endeavoured to improve this scheme, we follow this direction with following novelties:
>
> * **Novel Flow-Based Generative Inpainting**:We introduce a robust system of **flow-based generative model for semantic inpainting**, providing superior fidelity and handling uncertainty.
> * **New Scheme for Distilling Task-Specific LLM Prior**:Our method uniquely distills **structured, task-relevant commonsense knowledge from LLMs offline**. This specialized prior is directly integrated into our generative model, enabling it to **operate efficiently during inference without requiring any LLM calls**.
>
> # Weakness2: Effectiveness of "Distilling LLM Prior".
> We direct reviewer's attention to Table 3, where the improvement from incorporating the LLM prior is much higher in the in-domain setting, **increasing SR from 38.8 to 41.7**. This represents a substantial 2.9 absolute gain, clearly demonstrating the effectiveness of this module in enhancing performance. While the benefits might be slightly attenuated in the highly challenging cross-dataset transfer setting (Table 1), the in-domain results strongly validate the module's contribution.
>
> We appreciate you highlighting the challenge of prompting LLMs for complex co-occurrence patterns. While prompting with only a central object might indeed fall short in capturing the richer context of clustered objects, we **address this in complex scenarios by employing a practical data augmentation trick**: during training data generation, **we prioritize augmentating  objects that are part of co-occurring sets identified by more than one observed objects (namely the intersection of co-occuring sets of observed objects).** We initially omit this detail as it's simplicity, but we will include it in appendix in the final version for better reproductivity. We hope this can to some degree solve the problem in complex scenario without extremely more prompts. (Say if we consider the probability of cluster with 3 objects, there will be $\binom{21}{3} = 1330$ possible pairs, making it exhausting)
>
> # Weakness3: Effectiveness of flow model with "Dependent couplings".
>
> We clarify as follows:
>
> * First, in challenging real-world benchmarks like MP3D, **a 2.5 absolute gain in SR is substantial, not marginal**, especially from an already strong baseline (SR: 39.0). For context, prior works such as SG-Nav [5] report a 2.1 SR gain by incorporating complex scene graph relationships, and similar conclusions can be drawn from PONI's [4] ablations on MP3D. This demonstrates that our observed improvement is significant within this domain.
>
> * Second, our primary motivation for introducing dependent couplings was to **reduce computational complexity and significantly improve inference time** (as the reviewer also notes, from 17.13ms to 12.38ms). The observed performance gain was **an unexpected and welcome side-product**, highlighting the dual benefits of this design choice in achieving both efficiency and enhanced navigation performance.
>
> * Third, the hyper-parameters tuned in Appendix D.1 (number of steps $n$ and expansion ratio $\epsilon$) are fundamental to the operation of our generative flow model itself, irrespective of whether independent or dependent couplings are used. For fair comparison, all the hyper-parameters are kept the same when testing the effectiveness of dependent couplings.
>
> # Weakness4&Question3: Interpretation for hyper-parameters tuning curve.
>
> We appreciate the reviewer's careful review of our hyper-parameter analysis.
>
> First, we clarify that performance **does not fluctuate significantly with hyper-parameter settings**. We apologize if Figures 4 and 5 in the Appendix convey a misleading impression of drastic changes due to their scaling. In absolute terms, the maximum SR difference is less than 2.0 for the number of steps ($n$) and less than 1.5 for the expansion ratio ($\epsilon$). We will update these figures to better illustrate the absolute performance changes. Moreover, we will analyze that **$n$ is not about extensive tuning and generally increase it makes thing good.**
>
> Next, we provide the following interpretations for effective tuning:
>
> * Number of Steps ($n$): As n controls the discretization of the Ordinary Differential Equation (ODE) solver, **performance generally improves with increasing $n$ due to enhanced precision**. However, this gain rapidly saturates, becoming negligible against inherent navigation randomness. Thus, **tuning $n$ is primarily a trade-off to find the saturation point that balances performance with computational cost.**
>
> * Expansion Ratio ($\epsilon$): As detailed in lines 206-208, $\epsilon$ scales the influence of the LLM-derived prior on the generative process. **A moderate value (e.g., 1.2-1.3) typically yields strong results**. For LLMs that demonstrate higher reliability in inferring relationships for more distant objects (e.g., a TV being opposite a sofa may exhibit a large distance but also with large confidence), increasing $\epsilon$ can provide additional performance benefits, as it allows the model to leverage this "distant but strong" relational knowledge more effectively.
>
> Finally, we emphasize that GOAL is **generally not highly sensitive to hyper-parameters**. As shown in Table 7, many parameters were selected intuitively without extensive tuning, yet the model performs robustly. Our focus on tuning $n$ and $\epsilon$  was partly due to their computational efficiency, as they only require re-evaluation, not full model retraining.
> # Weakness5&Question1: Performance with models on consumer-grade GPU.
>
> We first stress that, in our setting, whether the LLM can be run on consumer-grade GPU does not count, as we only need to prompt API at the very beginning, which **cost less than 0.3$ for ChatGPT4**, and **without any memory comsumption brought by local deployment**.
>
> However, we understand your curiosity regarding the performance of GOAL with a smaller LLM. We are currently conducting experiments with Llama3:8b (we noted that SG-Nav, while stating LLaMA 7B in their paper, uses Llama3.2-vision:11b according to their official GitHub repository). We will provide these results to you within the discussion period as soon as the experiments are completed.
> # Question2: Gap of content generated by different LLMs
> **We evaluate quantatively of the gap of content from different LLMs, the results show there is difference, but in small scale.**
> To assess the variability in distilled knowledge across different Large Language Models, we measured the difference in content between the generated distance and confidence matrices using the following metrics:
>
> 1. **Consine Similarity** measures if two matrices have similar patterns, that is, if they always agree a spcific object pair to have a higher/lower distance/confidence:
> $
> \text{Cosine}(A,B) = \frac{A\cdot B}{||A|| ||B||}.
> $
>
> 1. **Average Frobenius Norm** measures the average absolute differences of different results:
> $
> \text{AverageFrobenius}(A,B) = \frac{1}{N_C^2}\sqrt{\sum_{i=1}^{N_c}\sum_{j=1}^{N_c}|A_{ij} - B_{ij}|^2}.
> $
>
> The computed similarity scores are as follows (each element scheduled as distances / confidences)
>
> |LLMs pairs|Consine|Frobenius|
> |---|---|---|
> |ChatGPT-ChatGLM|0.947/0.964|0.062/0.0092|
> |ChatGPT-DeepSeek|0.956/0.960|0.057/0.0085|
> |ChatGLM-DeepSeek|0.933/0.954|0.071/0.0103|
> |Llama-ChatGLM|0.952/0.952|0.059/0.0115|
> |Llama-ChatGPT|0.921/0.952|0.074/0.0095|
> |Llama-Deepseek|0.892/0.938|0.088/0.0104|
>
> From these results, we observe that:
>
> * Among the industry-level models (ChatGPT, ChatGLM, DeepSeek), ChatGLM exhibits slightly more distinct content compared to the other two.
> * The consumer-grade model, Llama, shows comparatively larger differences when paired with other models.
> * Despite these variations, all LLM pairs **demonstrate high cosine similarities** (ranging from 0.892 to 0.956 for distances and 0.938 to 0.964 for confidences). This indicates that all models consistently **output fundamentally similar spatial relationship patterns**.
> * The absolute differences, as measured by the Average Frobenius Norm, **typically range from 5cm to 10cm for distances and around 1% for confidences**.
>
> Since we formulate the LLMs' responses as smooth Gaussian fields, they are expected to be robust to slight differences.
>
> # Question4: Per-episode inference time reduction for dependent couplings.
>
> We thank the reviewer for suggesting this more practical metric. We analyzed the average per-episode inference time, accounting for varied scene and goal object types.
>
> Since inference time of an episode varies according to various factors, we take the first 20 episodes of all 11 scenes (totally 220 episodes) and compute the average inference time:
>
> |Data Couplings|Inference time per episodes|
> |---|--------------|
> |Independent Couplings (w cross-attention)|3m 43s|
> |Dependet Couplings|2 m 30s|
>
> This represents **a significant 32.7% reduction in average episode inference time (from 223s to 150s)**, demonstrating the practical efficiency benefits of dependent couplings beyond individual forward passes.
>
> We hope these comprehensive responses and additional analyses address all your concerns.
>
> [1] Learning to Map for Active Semantic Goal Navigation. ICLR 2022.
>
> [2] SSCNav: Confidence-Aware Semantic Scene Completion for Visual Semantic Navigation. ICRA 2021.
>
> [3] Imagine Before Go: Self-Supervised Generative Map for Object Goal Navigation. CVPR 2024.
>
> [4] PONI: Potential Functions for ObjectGoal Navigation with Interaction-free Learning.CVPR 2022.
>
> [5] SG-Nav: Online 3D Scene Graph Prompting for LLM-based Zero-shot Object Navigation. NeurIPS 2024.

---

> > ### Author Response · Authors · 2025-08-01
> > **Updated Results with Llama3:8b**
> >
> > We apologize for the delay of requested additional experiment in response to Weakness5&Question1. Thank you for your suggestion, and we have now completed the experiments with Llama3:8b. The quantitative results, show that there is **approximately a 1 SR decrease** compared with the more powerful ChatGPT:
> >
> > |LLM| SR| SPL| DTS|
> > |------|----|------|------|
> > |Llama3| 40.6 | 15.1| 4.86|
> > |ChatGPT| 41.7|15.5|4.84|
> >
> > This demonstrates that while there is a slight performance drop, our framework **remains effective even with a smaller, more efficient model**.

---

### Official Review · Reviewer_wb79 · 2025-07-03

**Clarity:** 3
**Significance:** 3
**Originality:** 3
**Rating:** 4
**Confidence:** 4

**Summary:**

This work presents a generative flow-based framework that models the semantic distribution of indoor environments by bridging observed regions with LLM-enriched full-scene semantic maps.

**Questions:**

1. Does the LLM give the same numbers for the distance and confidence, even when the same prompt is used as the input of the LLM?
2. In Figure 8, the gym equipment and fireplace seem to be in multiple, distant places in the same environment. How accurate are the generated maps? Is there a numerical method to measure this?

**Ethical Concerns:**

["NO or VERY MINOR ethics concerns only"]

**Final Justification:**

The authors have addressed all the concerns that I raised.

**Limitations:**

I am not sure how reliable are the numerical values of distance and confidence that are an output of LLM. As the map is generated using this information from the LLM output, I am unsure how reliable is the quality of generated maps.

**Paper Formatting Concerns:**

There are no major formatting concerns.

**Quality:**

3

**Strengths And Weaknesses:**

The authors show state-of-the-art performance on Gibson and MP3D datasets. Moreover, they show good generalisation on HM3D dataset.

The authors have not compared the performance of their model with RIM [1].

The generated maps rely on an LLM prompt to predict numerical values of distance and confidence, alongwith reasoning. I think there would be some randomness in the output of values for distance and confidence. As LLMs are autoregressive models, I am not sure how reliable they would be at predicting numerical values, also given that they are prone to hallucination. Perhaps, you could train a transformer encoder to predict the values of distance and confidence, instead of prompting an LLM.
Moreover, the qualitative results in the appendix in Figure 8 show gym equipment and fireplace in multiple, distant places in the same environment. Usually in a house, all the gym equipment is present in one area and not scattered in different places. So, I am not sure how accurate and reliable are generated maps.

[1] Chen, Shizhe, Thomas Chabal, Ivan Laptev, and Cordelia Schmid. "Object goal navigation with recursive implicit maps." In 2023 IEEE/RSJ International Conference on Intelligent Robots and Systems (IROS), pp. 7089-7096. IEEE, 2023.

---

> ### Author Rebuttal · Authors · 2025-07-31
>
> We appreciate your insightful questions and concerns, which we address comprehensively below.
>
> # Weakness1: Comparison with RIM
> We appreciate you raising the question of comparison with RIM [1]. While RIM is an interesting and important work in object goal navigation, we did not include a direct empirical comparison for the following reasons:
> * RIM involves **extensive data collection through robot interaction in simulators and incorporates auxiliary training tasks**, as also noted by T-Diff [2]. This makes a direct and fair comparison challenging.
> * RIM **exhibits a fundamentally different paradigm from our work**. It does not incrementally build a map to map on, nor incorporating LLM knowledge. We mainly compared with methods more aligned with our explicit map-centric and LLM-enhanced approach (e.g., UniGoal, GAMap, SG-Nav, T-Diff, SGM).
>
> However, we agree on RIM's relevance within the broader object goal navigation landscape. We will ensure it is properly cited and discussed in the "Related Work" section of the revised manuscript to contextualize our contributions more comprehensively within the field.
>
> # Weakness2&Question1: The randomness of LLM response.
>
> We acknowledge the reviewer's valid concern regarding the inherent variability in LLM responses, even with identical prompts. **We provide quantitative results to demonstrate that the differences are minor.** To quantitatively assess the consistency of our distilled LLM prior, we measured the similarity between multiple distance and confidence matrices generated by the same LLM (ChatGPT) using the same prompt.
>
> We employed two key metrics:
>
> * **Cosine Similarity**: Measures the angular similarity between two matrices, indicating if they share similar patterns of higher/lower distances/confidences for object pairs:
> $
> \text{Cosine}(A,B) = \frac{A\cdot B}{||A|| ||B||}.$
> * **Average Frobenius Norm**: Quantifies the average absolute element-wise difference between two matrices, providing a direct measure of numerical divergence:
> $
> \text{AverageFrobenius}(A,B) = \frac{1}{N_C^2}\sqrt{\sum_{i=1}^{N_c}\sum_{j=1}^{N_c}|A_{ij} - B_{ij}|^2}.$
>
> **Methodology**: We prompted ChatGPT five times with the same input. Treating the first result as a reference (A), we computed these two metrics between the reference and the subsequent four results (B1-B4).
>
> **Results**: The consistency analysis is summarized below (values for Distance Matrices / Confidence Matrices):
>
> ||Cosine$\uparrow$|Frobenius$\downarrow$|
> |----|----|-----|
> |B1|0.976/0.992|0.048/0.0040|
> |B2|0.968/0.988|0.050/0.0047|
> |B3|0.981/0.991|0.039/0.0041|
> |B4|0.974/0.990|0.047/0.0045|
> |Average|0.975/0.990|0.046/0.00433|
>
>
> This analysis clearly demonstrates that responses from the same LLM, with identical prompts, **exhibit highly similar patterns** (average Cosine Similarity of 0.975 for distance and 0.990 for confidence). Furthermore, the **absolute differences are very small** (average Frobenius Norm indicating differences typically less than 5cm for distances and 0.5% for confidence values). Given that our semantic maps are represented as relatively smooth Gaussian fields, these small numerical differences are not expected to cause significant performance degradation. The generative model effectively learns to interpret this consistent, albeit slightly variable, prior for robust semantic scene completion.
>
> # Weakness3: Hallucination of LLMs.
>
> We acknowledge the reviewer's concern regarding LLM hallucination, a known characteristic. Our approach **specifically mitigates this through a comprehensive, hierarchical prompting strategy designed to guide structured reasoning and constrain the output space**, thereby significantly reducing instances of hallucination in the distilled prior.
>
> We strongly contend that **despite their limitations, the commonsense reasoning capabilities of LLMs are immensely valuable for ObjectNav tasks**, particularly for generalizing to unseen environments and novel object categories. This perspective is increasingly shared within the community, with a growing number of recent works demonstrating effective LLM applications in ObjectNav, including L3MVN [2], SGM [6], VoroNav [5], GAMap [4], SG-Nav [7], and UniGoal [3] etc. These works, alongside ours, highlight that judicious application of LLMs can significantly enhance embodied agents' capabilities, and the concern of hallucination does not warrant discarding their utility entirely for this domain.
>
> # Weakness4: Potential method for a transformer encoder instead of LLMs.
>
> We thank the reviewer for this constructive suggestion regarding a transformer encoder. However, we stress that **a transformer encoder trained on existing datasets will not be comparable to the LLM which is trained with web-scale data.** An encoder trained on existing scene datasets would inherently be limited to the in-domain knowledge present within those specific datasets. Such an approach would essentially regress to patterns already observable in current scene data, offering **no fundamental difference from directly predicting semantic maps** trained with solely existing scene datasets.
>
> Our core motivation for augmenting semantic maps with LLMs is precisely to **leverage their vast commonsense knowledge, derived from web-scale data, which is crucial for out-of-domain generalization**. This allows our model to "imagine" plausible semantic layouts for unseen environments and novel object categories – a capability that a transformer encoder trained purely on existing scene datasets, without equivalent pre-training on diverse world knowledge, would not possess. Therefore, the LLM serves as a unique and invaluable source of generalized world knowledge for our task.
>
> # Weakness5&Question2: Misunderstanding on qualitative results.
>
> We appreciate the reviewer's close inspection of our qualitative results in Appendix Figure 8.
>
> First, we wish to clarify a misunderstanding: **the objects identified by the reviewer as "gym equipment" are actually "seating"**. We ask the reviewer to further check and identify the figure carefully.
>
> Based on this clarification, we justify the qualitative results as follows:
>
> * **"Seating" Placement**: The scattering of seating across a scene, as observed, is quite common in diverse indoor environments (e.g., multiple living areas, dining nooks, individual chairs in hallways). In Figure 8, the generated seating instances are consistently placed in reasonable contexts, such as near walls with pictures or alongside generated tables, reflecting plausible arrangements.
> * **"Fireplace" Placement**: The reviewer noted "multiple, distant" fireplaces. Upon careful re-inspection of Figure 8, only the second row depicts two fireplaces, each placed behind a wall with pictures and accompanied by generated seating. This configuration is a common and reasonable arrangement for distinct living areas within a larger residence.
>
> * **Purpose of GOAL**: The purpose of GOAL is not to reconstruct exactly the whole scene semantic, but **focus more on plausible objects placement to guide object navigation**. This is the reason why we try to distill the object co-occurence information into flow model, for our object-level task.
> # Question2: Numerical method to measure the quality of generated map
> We want to emphasize that **measuring the quality of the generated map quantitatively via numerical metrics can be misleading because our focus is on the downstream object navigation task**, and the performance on this task (SR, SPL, DTS) is the most fair and ultimate judgment. Our generative model produces plausible semantic completions, not exact reconstructions of a hidden ground truth, and these plausible completions are what enable robust navigation.
>
> However, since the reviewer is curious about relevant information, we have conducted an evaluation using metrics commonly used in segmentation tasks: mIoU, mAcc (on all 21 objects and free space, totaling 22 categories), and mean squared error (MSE). It is important to note that we generate validation data samples similarly to training samples, but without adding the LLM prior, as the LLM prior is introduced with stochasticity, making direct ground truth comparison for the augmented maps less suitable for validation. We also test the performance of the flow model without LLM knowledge distilled for a fairer comparison, as the supervision for the flow model with LLMs is no longer identical to the expected whole scene semantic map, but instead incorporates strong object co-occurrence information.
>
> We compare the validation results with a recent work, SGM (also without LLM knowledge for fair comparison), in the same setting:
>
> |Model|mIoU$\uparrow$|mAcc$\uparrow$|MSE$\downarrow$|
> |------|-----|------|------|
> |SGM|11.05| 12.70| 16.18|
> |GOAL| **35.20**|**35.63**|0.013|
>
> Note that SGM adopt a MAE-like algorithm, which will normalize the input and the MSE may not comparable, we refer the reviewer only to our absolute MSE instead of comparing with SGM.
>
> We hope these comprehensive responses and additional analyses address all your concerns.
>
> [1] Object goal navigation with recursive implicit maps. IROS 2023.
>
> [2] L3MVN: Leveraging Large Language Models for Visual Target Navigation. ICRA 2023.
>
> [3] UniGoal: Navigate to Any Goal in Zero-shot! CVPR 2025.
>
> [4] GAMap: Zero-Shot Object Goal Navigation with Multi-Scale Geometric-Affordance Guidance. NeurIPS 2024.
>
> [5] VoroNav: Voronoi-based Zero-shot Object Navigation with Large Language Model. ICML 2024
>
> [6] Imagine Before Go: Self-Supervised Generative Map for Object Goal Navigation. CVPR 2024.
>
> [7] SG-Nav: Online 3D Scene Graph Prompting for LLM-based Zero-shot Object Navigation. NeurIPS 2024.

---

> > ### Comment · Reviewer_wb79 · 2025-08-04
> >
> > As the authors have addressed all my concerns, I'll be raising my score.

---

> > > ### Author Response · Authors · 2025-08-04
> > >
> > > We thank the reviewer for the positive feedback. We will continue to improve our work.

---

### Official Review · Reviewer_Hp6p · 2025-07-05

**Clarity:** 2
**Significance:** 3
**Originality:** 2
**Rating:** 5
**Confidence:** 4

**Summary:**

This paper introduces a framework for ObjectNav based on semantic map completion with a flow-matching-based generative model. It uses an LLM to perform a form of data-augmentation, injecting objects into semantic maps during training based on a pre-trained LLM’s prediction about likely co-occurring objects. At test time, the flow-matching generative model completes a plausible semantic map of the scene given the history of observations, which is then used to select navigation targets for ObjectNav. The method is compared against several baselines for navigation via envrionment imagination, showing SOTA and transfer to environments that the generative model is not trained on.

**Questions:**

Based on equations 6,7,8 it seems as if there is no protection for augmenting objects being placed across a wall from it’s co-occurring object, making it’s geodesic distance much farther than the predicted distance.

It’s not clear to me exactly how segmentation is performed. As is it seems as if this work trained a sparse convolutional network from scratch on RGB pointclouds. If this is is the case there should be more details included about training.

**Ethical Concerns:**

["NO or VERY MINOR ethics concerns only"]

**Final Justification:**

As authors have addressed my questions and concerns, I change my score to accept.

**Limitations:**

Yes

**Quality:**

3

**Strengths And Weaknesses:**

*Strengths*
- This paper proposes a strong system for object navigation with several good ideas and engineering choices. Their LLM based map-augmentation scheme incorporates priors from LLMs effectively while avoiding the need for running an LLM at inference time, and the use of flow matching for map inpainting is a natural upgrade in methodology for this line of work based on more current inpainting research.
- Paper is well written and the language is clear. Many related works are compared against and many relevant ablations are included.
- The results are strong and clearly give SOTA performance.


*Weaknesses*
- Missing details on the “Effectiveness of data-dependent couplings” ablation: Inpainting/outpainting with diffusion models is already a well studied problem. While the inpainting formulation used in this paper is compelling and could make sense, there are very few details about the baseline used to justify the contribution of this non-standard inpainting scheme. UNet style models typically concatenate the conditioning to noise [1] while several different conditioning schemes are used for DiT based models [2,3]. This paper would be stronger if these methods were explicitly compared against, or one of these inpainting schemes were adopted directly. As is, there is not enough to justify the proposed inpainting scheme as a contribution.

- Incomplete ablations against SGM. Some of the major differences between this work and SGM that need more experimentation:
1) Scene segmentation vs per-frame segmentation
2) The use of flow matching vs MAE
3) LLM conditioning techniques (conditioning on features vs directly augmenting maps)
SGM claims to do segmentation using single frame models (Mask-RCNN, RedNet) while this work uses scene segmentation. In Table 3 this work reports a 5 point SR increase from using scene segmentation. If the same increase were found from simply using scene segmentation with SGM, then SGM would actually be out-performing this method on MP3D, implying that either MAE or the SGM language conditioning technique is superior to the techniques from this paper. This paper would benefit from a comparison against SGM using scene segmentation, and SGM using this method’s language augmented maps as part of training.

[1] Rombach, Robin, et al. "High-resolution image synthesis with latent diffusion models." Proceedings of the IEEE/CVF conference on computer vision and pattern recognition. 2022.

[2] Nitzan, Yotam, et al. "Lazy diffusion transformer for interactive image editing." European Conference on Computer Vision. Cham: Springer Nature Switzerland, 2024.
[3] Wu, Xian, and Chang Liu. "DiTPainter: Efficient Video Inpainting with Diffusion Transformers." arXiv preprint arXiv:2504.15661 (2025).

---

> ### Author Rebuttal · Authors · 2025-07-31
>
> We sincerely thank you for your comprehensive review and valuable feedback on our submission. We are pleased that you recognized the strengths of our work, including the novel LLM-based map augmentation, the effective use of flow matching for map inpainting, the well-written paper, and the strong, state-of-the-art results. We address your concerns below.
>
> # Weakness1: Missing details and potential baselines for 'Data-dependet coupling'
> ## Details of the 'Independent Couplings' Baseline
> We apologize for inclarity brought by our incomplete details for the baseline for 'independent couplings', only give a brief introduction in line 265-267. Here we first dive more into the conditional mechanism for baseline: We adopt **two convolution blocks to downsample the partial semantic map to avoid computational bottleneck**, and adopt a Patch embedder to project it into higher dimension. Then we interpolate Gaussian noise with ground truth map and also send into a patch embedder. We then use **DiT blocks with cross attention layers** [1], since we suppose the partial semantic map condition is highly expressive and simple layer norm may not suitable. The interpolated noisy patches are send into sequential DiT blocks as input, and partial semantic map patches are **sent to the cross attention layer of each DiT block to serve as condition**.
>
> ## Comparison with DiTPainter
>
> Since reviewer also mentioned some of other conditioning mechanism, we choosed to compare with DiTPainter [2] mentioned as it also adopt a flow matching training pipeline and reveal similar idea. It is important to note that **DiTPainter was preprinted on Arxiv only three weeks before the NeurIPS 2025 submission deadline, making both works concurrent in their exploration of direct coupling mechanisms**.
>
> As DiTPainter does not open-source their codes, we first reimplement it according to the main idea and adapt to our objectgoal task. It adopts a somehow similar mechanism with ours: They also adopt a dependent couplings mechanism but with more inputs. They project interpolated noise, corrupted video, and binary masks to higher dimensions individually, and add them to serve as the input to model, namely $x = \text{embedder}_1(x_0) + \text{embedder}_2((1-t)\epsilon + tx_1) + \text{embedder}_3(M)$ where $x_0$ is the corrupted data, $x_1$ is the full videos, and $M$ is the mask representing corrupted regions. However, we merge the information of multiple inputs before projection, namely $x = \text{embedder}(f(x_0, x_1, \epsilon, M))$ which eliminate the cost of extra embedders.
> |model| SR$\uparrow$| SPL$\uparrow$ | DTS$\downarrow$
> |-------| ------|--------|-------|
> |DiTPainter|41.4| **15.6**| **4.84**|
> |GOAL| **41.7** | 15.5 |**4.84**|
>
> As shown, GOAL achieves comparable navigation performance with slightly higher SR, while incurring less model complexity due to our streamlined input processing. Given the comparable performance and reduced complexity, we will retain our conditioning mechanism. While our work is concurrent with DiTPainter, we acknowledge that this specific conditioning mechanism should be presented as a technical improvement for building a strong flow-matching system for object-goal navigation rather than a core conceptual contribution in the revised version of the paper.
>
> # Weakness2: Incomplete comparison with SGM
> We conduct multiple comparison with SGM. The experimental results shows that **SGM benefit less from scene segmentation**, inheriting from the random masking scheme of MAE. Furthermore, **flow matching exhibits stronger capabilities of generalization in transfer setting**.
>
> Since SGM does not open-source their models and evaluation scirpts on MP3D, we reproduce it in reference to their codes for training and evluating SGM on Gibson. However, as they also did not open source the exact prompts and details on how they get language guidance, we reproduce only the naive version without language guided, and refer it to MAE. To reach a fair comparison, we compare MAE with our method wihout LLM, referring to FM.
> ## Scene segmentation vs per-frame segmention
> we refer scene segmentation to 'SG'
>
> |Algorithm| SR$\uparrow$| SPL$\uparrow$|DTS$\downarrow$
> |---------|-----| ------| -----
> | **w/o SG** ||||
> |MAE| 31.9 |**11.7**|5.31
> |FM|**32.4**|**11.7**|**5.25**
> | **w/ SG** ||||
> |MAE+SG|36.7|14.3|5.16
> |FM + SG|**38.8**|**14.6**|**5.08**
>
> Our method delivers stronger performance in both settings, with more performance gains in the w/ SG configuration. This is largely because our direct coupling method in FM relies more heavily on the reliability and consistency of the entire observed semantic map, which scene segmentation enhances. For MAE, which typically selects a fixed number of patches as input, the inherent consistency of the input semantic map can be disrupted by the patch selection process, thus limiting the benefit of consistent scene segmentation.
>
> ## Flow matching vs MAE
> We further test the capabilities of generalization in transfer setting of Flow matching and MAE (trained on MP3D and directly test in HM3D), the results are as follows:
>
> |Algorithm| SR$\uparrow$| SPL$\uparrow$| DTS$\downarrow$|
> |----|-----|-----|-----|
> |MAE|32.1|**14.70**|4.85|
> |FM|**35.9**|14.69|**4.65**|
>
> FM achieve a 3.8 SR gains, and the reason of minor decrease of SPL (0.01) has already discussed in the main paper. **Flow matching exhibits a significant better generalizability probably due to its multimodal output**,  which better capture the underlying distribution of possible semantic maps in unseen environments.
>
> ## LLM conditioning techniques.
> We acknowledge that directly transferring our LLM augmentation mechanism to MAE is challenging, as MAE's training paradigm (masking random patches) does not inherently involve the concept of partial and complete maps in the same way our flow-based model does. Our augmentation mechanism is specifically designed to simulate real navigation paths and augment only at the boundaries of observed regions.
>
> Moreover, our primary goal with LLM conditioning was **to completely eliminate the need for LLM usage during inference while still benefiting from its commonsense knowledg**e. We did not aim to surpass the language guidance in SGM, which operates on a different principle by directly conditioning on language features during inference. Our contribution lies in distilling this knowledge offline into a generative model that can then operate independently.
>
> # Question1: Protection for placing objects across a wall
>
> We appreciate you raising this important practical consideration. While our LLM-distilled prior provides commonsense spatial relationships based on **Euclidean distances** (instead of geodesic), our modified local planner (initially designed for efficient inference) planning **according to FMM distance (geodesic)** can mitigate the problem during actual navigation.
>
> As stated in line 218, our modified local planner **selects a new long-term goal when the FMM distance of current goal is too far away**. If GOAL selects a goal across the wall, **the agent's planner would immediately detect that the geodesic distance is excessively large and prompt agent to plan again**. This makes the issue not critical for the overall navigation performance.
>
> # Question2: Details for segmentation.
>
> Yes, we confirm that we trained a sparse convolutional network from scratch on aggregated RGB point clouds for 3D semantic segmentation. We apologize for the oversight in not directly referencing the detailed training specifications in the main paper.
>
> **These crucial details regarding the segmentation model's training can be found in Appendix C.5**. We will ensure that a clear and prominent reference to Appendix C.5 is added to the main paper in the revised version to enhance reproducibility and clarity for the readers.
>
> We hope these clarifications and additional results address your concerns. We are confident that with these improvements, the paper will be even stronger.
>
> [1] Scalable Diffusion Models with Transformers. ICCV 2023.
>
> [2] DiTPainter: Efficient Video Inpainting with Diffusion Transformers. arixv preprint 2025.

---

> > ### Comment · Reviewer_Hp6p · 2025-08-04
> >
> > Thank you for the author's thorough response. I appreciate the effort to compare with baselienes, despite not having access to open sourced code or models.
> > - I agree that given that the proposed mechanism has comparable performance and similar complexity, it is reasonable to include the newly proposed mechanism as a contribution. I would suggest to include the comparison from the rebuttal in the revised paper's appendix to justify the design.
> > - The comparisons with SGM add more clarity into the strengths of the proposed method.
> > - The rest of my questions have been clarified.
> >
> > Given the above, I change my rating to accept.

---

> > > ### Author Response · Authors · 2025-08-05
> > >
> > > We thank the reviewer for the positive feedback. We will continue to improve our work.

---

### Decision · Program_Chairs · 2025-09-17

**Decision:**

Accept (poster)

**Comment:**

The paper proposes to use a generative model leveraging the prior encoded in LLMs to complete the map of the scenes. It received two Accept and two Borderline Accept ratings. The reviewers mentioned various strengths of the paper, such as:
- Achieving strong SoTA performance on the object navigation task.
- Clear writing and illustrations.
- Notable features, such as eliminating the need to run an LLM during inference.
- Useful ablation experiments

The rebuttal addressed the concerns well (e.g., incomplete ablations wrt SGM and the randomness of LLM outputs). The AC checked the paper, the reviews, and the rebuttal. The AC concurred with the recommendation of the reviewers and recommended acceptance.